# Breeding New Premium Quality Cultivars by Citrus Breeding 2.0 in Japan: An Integrative Approach Suggested by Genealogy

Tokurou Shimizu 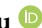

Division of Citrus Research, Institute of Fruit Tree and Tea Science, National Agriculture and Food Research Organization, Shimizu, Shizuoka 424-0292, Japan; tshimizu@affrc.go.jp

**Abstract:** Developing varieties with diverse features that satisfy varied commercial needs, improving overall fruit quality, and quickly releasing them, are prerequisites in citrus breeding. However, these three goals require trade-offs in conventional breeding, even with the application of the marker-assisted selection technique. Conventional breeding cannot achieve these three goals simultaneously and it has been regarded as a breeding trilemma. Integrating a genomics-assisted breeding (GAB) approach that relies on quantitative trait locus detection by genome-wide association study and genome-wide prediction of a trait by genomic selection using enriched marker genotypes enhances breeding efficiency and contributes to eliminating the trilemma. Besides these efforts, the analysis of the genealogy of indigenous citrus varieties revealed that many high-quality indigenous varieties were selected within a few generations. It suggested that selecting a new premium quality hybrid is possible by integrating it with the GAB technique and helps avoid the trilemma. This review describes how a new approach, "Citrus Breeding 2.0" works for rapidly developing new, premium quality hybrids and introduces three applications of this technique, specifically, rebreeding, complementary breeding, and mimic breeding based on the ongoing citrus breeding program in NARO, Japan.

**Keywords:** citrus breeding; marker-assisted selection; genomics-assisted breeding; genome-wide association study; genomic selection; genealogy

## 1. Introduction

Major citrus cultivars such as sweet orange, Clementine, lemon, grapefruit, and Satsuma have been produced worldwide for over 100 years, and are still considered economically important crops [1,2]. However, the demand has increased for high-quality new cultivars easy-to-peel and which meet consumers' needs in a short period [3]. Tremendous efforts have been devoted to mutation breeding in citrus to select a cultivar from sport, nucellar seedling [4–7], or after irradiation [8]. Though mutation breeding has successfully selected spontaneous or induced mutants in various varieties based on color break, rind or flesh color, fruit shape, acidity, or sugar content, mutation occurs at a low frequency [9], and the mutation spectrum depends on the varieties used for the method [3,5–7]. These limitations hamper the use of mutation breeding as a ubiquitous method for the development of new varieties. In contrast, crossbreeding is beneficial for developing novel, high-quality cultivars; however, in conventional breeding, it requires more time than mutation breeding, and the possibility to obtain a new elite hybrid depends on both the breeder's experience and chance.

Crossbreeding is a systematic approach for developing new cultivars with novel traits using hybridization [3,10,11]. However, it requires maintaining many seedlings in the orchard for an extended period and repeatedly evaluating their characteristics, and consequently this lengthy process is time-consuming and expensive. Therefore, in conventional breeding, satisfying fast breeding of a new hybrid, increasing fruit trait variation, and improving overall fruit quality are trade-offs, and it is difficult to achieve them simultaneously [12]. This constraint is regarded as a 'breeding trilemma' that is

difficult to avoid in conventional breeding [13]. To solve these constraints in conventional breeding, the marker-assisted selection (MAS) technique has been introduced in citrus breeding [12]. However, conventional MAS is limited to specific traits controlled by a few genes, and it is difficult to efficiently select various fruit traits that involve multiple genes in complex ways.

This review describes an ongoing comprehensive approach for integrating genomic-assisted breeding (GAB) with citrus genealogy on citrus breeding, called Citrus Breeding 2.0. This method can develop new cultivars with premium quality in a short period [13].

## 2. Citrus Crossbreeding Effort Aims to Improve Fruit Quality

### 2.1. Past Efforts of the Citrus Breeding Program in NARO

The fruit tree breeding program aims to develop new and unique high-quality cultivars. The citrus crossbreeding program of the Institute of Fruit Tree and Tea Science, NARO (NIFTS) started in 1946 and has produced hybrid cultivars intended for the domestic market [14]. To date, the program has developed 39 hybrid cultivars with eight intermediary mother selections by repeatedly selecting a superior hybrid from a single cross. When the breeding program started, Satsuma, sweet orange, ponkan mandarin, and pummelo were typical in Japan, but each cultivar had drawbacks. For instance, although Satsuma is easy-to-peel, seedless, and productive, it has less flavor and a lower sugar content than oranges and exhibits alternate bearing. Sweet orange sets fruits with high sugar content and a strong and excellent aroma that is suitable for direct consumption and juice production. However, the rind of the sweet orange is hard to peel and requires a knife to eat. Ponkan mandarin has a preferred flavor and higher sugar content than Satsuma, but it has a puffy rind and contains many seeds that hamper eating. Pummelo is the generic name of a set of cultivars that produce large fruit with a thick yellow rind. These pummelo sets fruits with a high soluble solid content (SSC) and pleasant aroma. But they are difficult to peel, set many seeds, and the segment membrane is difficult to chew. The initial goal of the program was to develop a new cultivar with peelability comparable with Satsuma and a preferable aroma similar to sweet orange. 'Kiyomi' is a tangor selected from a cross between Satsuma and a sweet orange cross [15]. 'Kiyomi' had better peelability than sweet orange, but not comparable to Satsuma, and while its aroma was superior to that of Satsuma, it was not comparable to sweet orange.

Besides its fruits, 'Kiyomi' has played a pivotal role in the breeding program, and it has been used as a mother plant because of its monoembryony (Figure 1). 'Kiyomi' has contributed to the development of ten hybrid cultivars ('Seihou', 'Tsunokaori', 'Youkou', 'Harumi', 'Akemi', 'Amaka', 'Nishinokaori', 'Tamami', 'Tsunonozomi', and 'Shiranuhi'), and six cultivars ('Amakusa', 'Setoka', 'Harehime', 'Reikou', 'Tsunokagayaki', and 'Asumi') are its progeny (Figure 1 and Table 1). These cultivars expanded the harvest season for five months (August to April) at open culture, enlarged fruit size (150–280 g), and improved SSC (>15). All of these cultivars except 'Amakusa', 'Tamami', and 'Tsunonozomi' set seedless fruit in open fields. Most of them are comparable to Satsuma in that they are easy to peel, and the segment membrane is easy to chew (Table 1). Many of these cultivars have "good" flavor; among which, 'Harehime', 'Tamami' and 'Asumi' have a strong and excellent aroma similar to that of sweet orange. In addition, some of these cultivars have unique features for less rind puffing ('Tsunokaori', 'Tsunonozomi'), less alternate bearing ('Amakusa'), grenadine to red rind ('Akemi', 'Reikou'), rich in beta-cryptoxanthin ('Tsunokagayaki', 'Asumi') and unique fruit shape ('Shiranuhi'). All these cultivars are protected under the Plant Variety Protection and Seed Act, Japan except 'Shiranuhi' [16], which is now produced in the USA ('Sumo citrus'), Korea ('Hallabong'), and Brazil ('Kinsei'). Thus, starting from 'Kiyomi', a series of cultivars that have diverse and improved traits have been bred through several crossings. It demonstrates the performance of crossbreeding that enables the expansion of diversity and improvement of fruit quality.

**Table 1.** List of the offspring cultivars of 'Kiyomi'. YTD: Years to development, SL: seedless (<5 seeds per fruit), BCP: beta-cryptoxanthin, SSC: soluble solid content.

| Cultivar | Year of Release | YTD | Harvest | Fruit Size (g) | Peelability | Seeds | Features |
|---|---|---|---|---|---|---|---|
| Kiyomi | 1979 | 30 | Late March | ~200 | Moderate | SL | A tangor (mandarin x orange). Orange-like aroma and juicy |
| Seihou | 1988 | 17 | Late January | ~200 | Hard | SL | Large fruit with orange-like good flavor and soft flesh |
| Tshunokaori | 1990 | 18 | March–April | ~160 | Easy | SL | High SSC (>13), and no rind-puffing |
| Amakusa | 1993 | 11 | August–January | ~200 | Hard | – | High SSC (>12), and less alternate bearing |
| Youkou | 1995 | 23 | January–February | 250–300 | Easy | SL | Large fruit with good flavor and soft flesh |
| Harumi | 1996 | 17 | January | 180–200 | Easy | SL | Large fruit with good flavor and soft flesh |
| Akemi | 1996 | 21 | March | 160–180 | Moderate | SL | High SSC (>12), grenadine rind, and soft flesh |
| Amaka | 1996 | 22 | August–January | 200–250 | Easy | SL | Soft fruit with good orange-like aroma |
| Nishinokaori | 1997 | 31 | August–January | 100–180 | Easy | SL | High SSC (>12), orange-like flavor, and soft flesh |
| Setoka | 1998 | 14 | February | 200–280 | Easy | SL | Large fruit with good flavor, high SSC (>12), and soft flesh |
| Harehime | 2001 | 11 | August | 180 | Easy | SL | Early type with good flavor, juicy, and soft flesh |
| Tamami | 2004 | 24 | January | 150 | Easy | – | High SSC (>12) and good orange-like flavor |
| Reikou | 2004 | 20 | January | 210 | Moderate | SL | High SSC (>12), good flavor, and red-skin |
| Tsunokagayaki | 2008 | 24 | January–February | 180–250 | Easy | SL | High SSC (>13), soft fruit, BCP rich, and suitable for greenhouse production |
| Tsunonozomi | 2011 | 37 | August | 190 | Easy | – | Early type with good flavor, high SSC (>12), juicy, and less rind puffing |
| Asumi | 2013 | 21 | February | 150 | Moderate | SL | Very high SSC (>15), orange-like flavor, and BCP rich |
| Shiranuhi * | | | February–March | 230 | Easy | SL | High SSC (>14), soft fruit, good flavor, and unique fruit shape |

* Not patented.

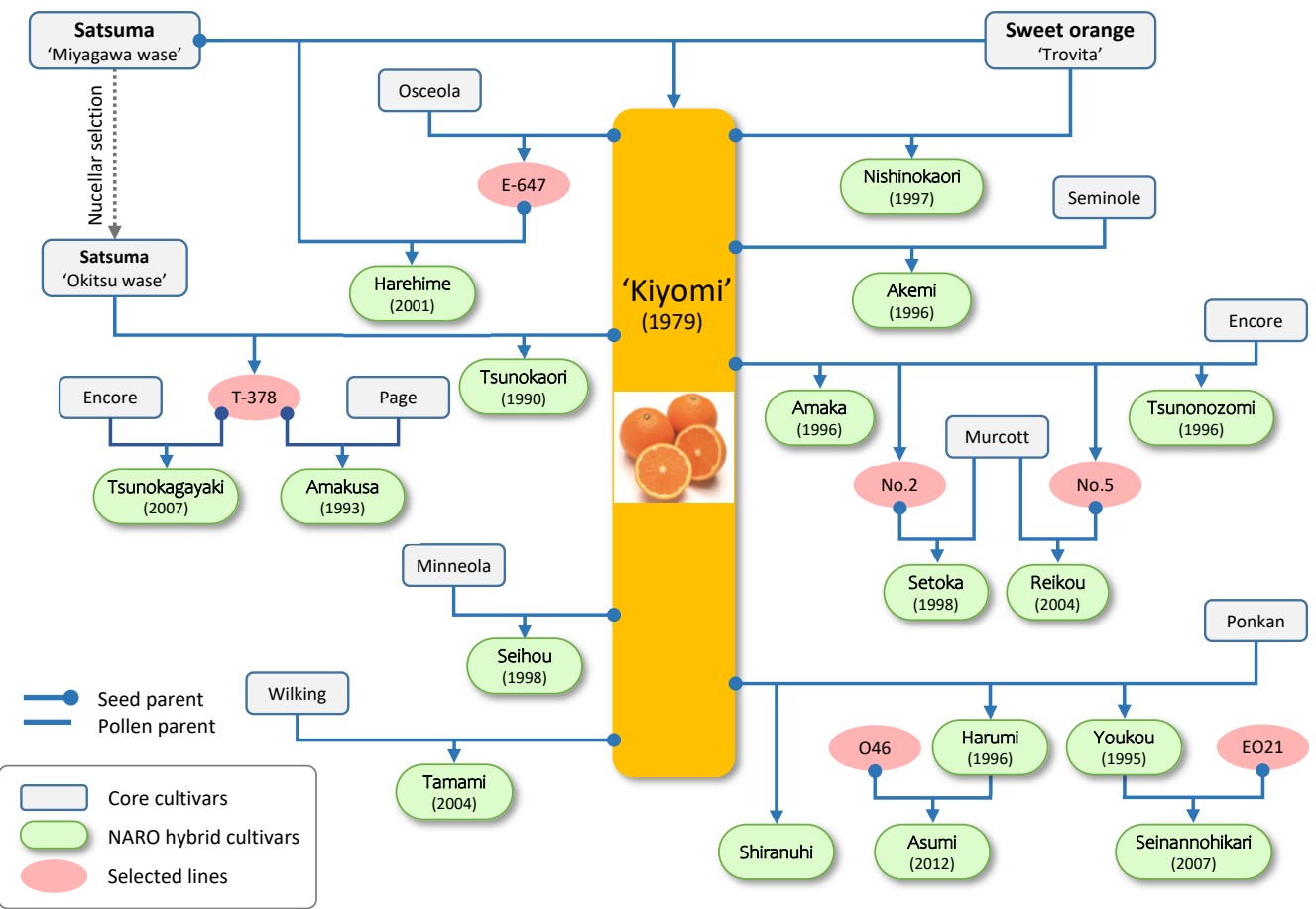

**Figure 1.** Pedigree tree of 'Kiyomi' tangor with its offspring cultivars. Connector line terminating in a circle denotes seed parent, straight end denotes pollen parent.

### 2.2. Constraints of Conventional Crossbreeding

Most fruit traits are susceptible to environmental variation. Therefore, conventional crossbreeding methods maintain many seedlings in the orchard for an extended period to evaluate their traits for several years to minimize the influence of potential variations. Alternate bearing is another drawback of some citrus fruits. Evaluating fruit traits at least twice is essential for reliable performance evaluation. However, it requires significant effort to maintain thousands of seedlings for the selection of just one elite hybrid. The NARO citrus breeding program took an average of 22.8 years to develop one promising selection from a cross (11–28 years) [13]. The breeding program provides approximately 5000 seeds each year for selection and then evaluates them for approximately 20 years; however, only one promising hybrid emerges every few years, and the rest are discarded. This means that even seedlings that will eventually be discarded must be maintained in the field for an extended period, and evaluation in the field is mandatory to determine which will be discarded and which will ultimately be selected. This step is time-consuming and incurs high costs over a long period to maintain the seedlings. Therefore, the cost of breeding a new cultivar can be reduced by increasing the number of promising hybrids acquired per seedling evaluated.

In the conventional breeding approach in NARO, the germinated seeds are nurtured for one to two years, after which a portion of them will be selected empirically for grafting at the orchard. The fruit traits of the seedlings selected for grafting are then evaluated for several years, and if any of them show promise, they will be subjected to further evaluation. Developing a high-quality new, unique hybrid is required to adjust for the broad array of consumer preferences [3,11]. In addition, there is now a need to breed new cultivars in the

shortest possible period to adjust for rapid changes in the market [12]. The citrus breeding program of NARO aims to achieve for satisfying three goals simultaneously: releasing new cultivars quickly, developing new cultivars that satisfy varied commercial needs, and improving overall fruit quality. However, satisfying these three objectives necessitates trade-offs—prioritizing one makes it difficult to achieve the others. Among those goals, the low probability of obtaining a promising hybrid from a set of the provided seedlings forms the background of the breeding trilemma. The insufficient number of candidate hybrids obtained from a single cross that causes generations needing to be repeated, prolonging the duration for breeding.

### 2.3. Pros and Cons of MAS for Improving Fruit Quality and the Demand for Genomic Breeding

Many of the seedlings provided for breeding will be discarded without evaluation at the initial step in conventional breeding processes. In the conventional breeding approach, the number of seedlings evaluated for their traits is identical to the number of seedlings grafted on the rootstock at the orchard. That means the number of seedlings that can be maintained in the field is the upper limit of the breeding size. The less elite individuals selected per seedlings maintained in the orchard are because many of the fruit traits are quantitative traits, and the occurrence of individuals exceeding the selection criteria is not high. The citrus breeding program of NARO evaluates over 20 fruit traits. Although some seedlings are excellent for several traits, but with few undesirable defects, they will not be selected as a candidate, which is another reason to decrease the probability of selecting the candidate.

If elite hybrids can be selected at the germinated seedling stage before grafting, and only these are grafted in the orchard and evaluated for fruit traits, the chance to select an elite hybrid can be improved (Figure 2) [12,17]. Using the MAS and Genomic selection (GS) to predict the fruit traits at the seedling stage before grafting makes it possible to discard unwanted seedlings before grafting. Considering the prediction accuracy of MAS or GS, this selection process does not guarantee that the selected seedlings should be above the threshold for the evaluated traits. However, these selection processes before grafting would enrich the portion of seedlings whose evaluated traits would exceed the selection threshold in the grafted population. Even if the number of seedlings maintained in the field is constant, using MAS and GS facilitates increasing the sum of seeds evaluated trait by orchard or predicted by MAS (virtual breeding size), providing a leverage effect. Thus, an increased chance of finding promising premium quality hybrids is anticipated.

The DNA marker selection technique is now used in a wide range of breeding programs [12,18]. Currently, DNA markers for the selection of the citrus tristeza virus (*Ctv*) resistance from *Poncirus trifoliata* [19], male sterility [20,21], and polyembryony [22] are available, and they are put in the citrus breeding program in NARO. The selection of zygotes from polyembryonic female parent using SSR marker analysis has also been conducted for many years [12]. Despite the benefit of MAS in breeding, most DNA markers and selection systems developed to date for MAS target traits are controlled by a single gene [12,17]. This is partially due to the difficulty in developing selection markers using quantitative trait locus analysis for citrus, which are large, have a long juvenile period, and require a long time to evaluate quantitative fruit traits in the field. In addition, owing to the higher genomic heterozygosity of citrus, segregation of target traits also relies on the cross combinations of the parents [12]. This further complicates estimating the locus position of the causal gene and developing linkage markers for selection.

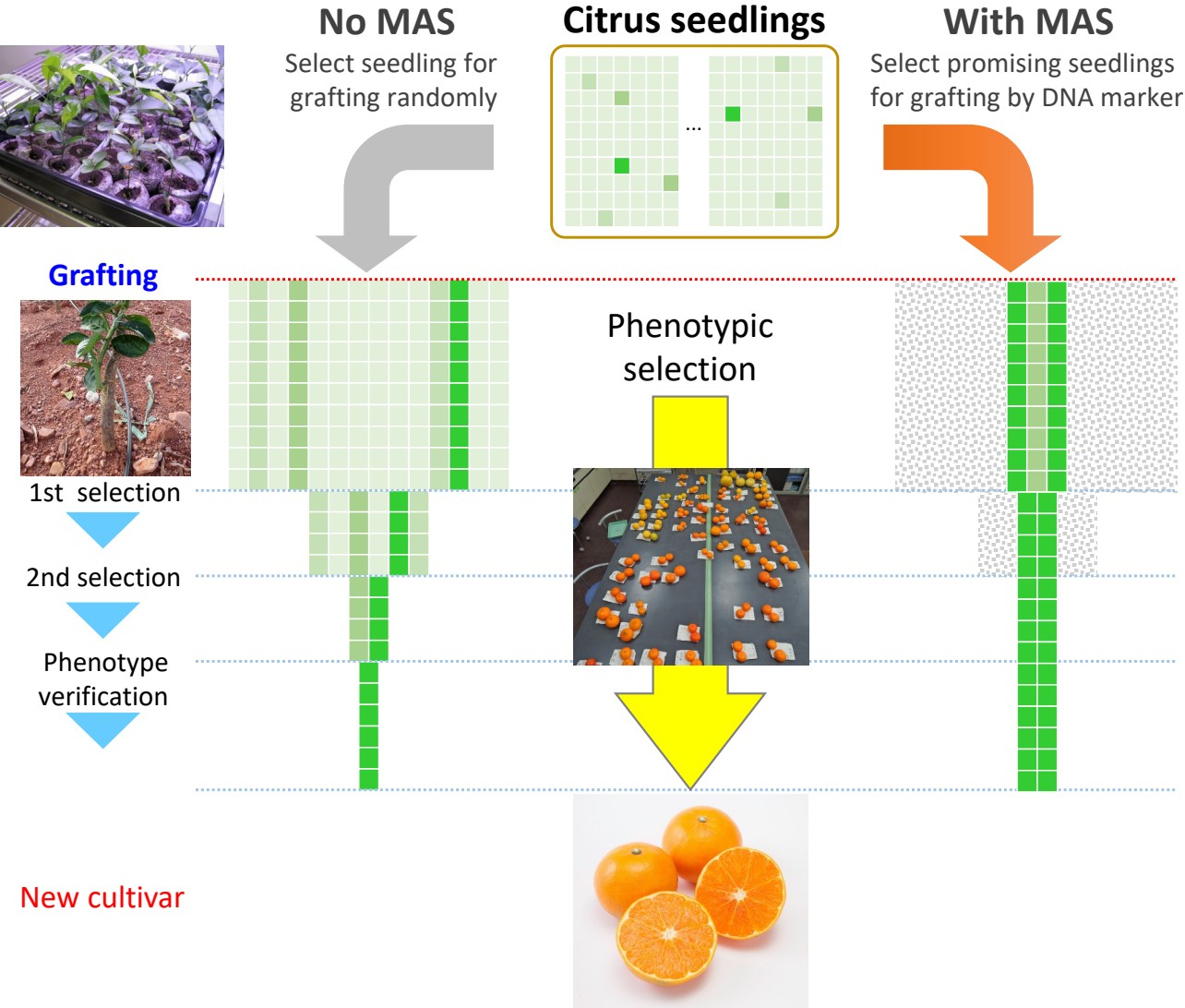

**Figure 2.** Scheme of how marker-assisted selection (MAS) works for improving breeding efficiency. Each square represents a seedling. A vivid green square corresponds to a promising seedling, and a pale one represents a seedling to be discarded. Conventional breeding that does not use MAS (left) selects citrus seedlings randomly then grafts them at the orchard; thus, the occurrence of elite hybrids is low. The breeding with MAS (right) selects candidate seedlings, then grafts the candidate seedlings, and presents a higher chance for selection of promising candidates than conventional breeding.

The Genome-Wide Association Study (GWAS) is an approach that detects linkage disequilibrium based on the genome-wide genotype of many native and bred cultivars and detects genomic regions strongly associated with a trait [23,24]. The regions detected by GWAS are the first candidates of the region of interest for developing DNA markers for MAS. The other approach, GS, performs genome-wide prediction using enriched marker genotypes. GS builds a prediction model for the trait of interest from genome-wide genotypes of broad samples with their phenotype data, and then predicts the traits of samples from genotype data using the prediction model [24–26]. In GWAS, preparing multiple segregating populations for linkage analysis and evaluation of their traits in the field is not required. Furthermore, by evaluating diverse cultivars, the problem of segregation of the trait of interest being affected by the population can be avoided. Although GS cannot identify the genomic region associated with the target trait, it can predict and select traits determined by many genes with minor effects [17,25–29]. In GWAS, increasing

the number of samples provided for linkage disequilibrium analysis and evaluating traits over multiple years to reduce errors effectively reduces false correlations [23]. Increasing genotype and trait data are also effective in improving the prediction accuracy of GS. The genomics-assisted breeding (GAB) approach, also called 'genomic breeding', that uses MAS based on GWAS and GS [24], is anticipated to break the constraints of conventional MAS, and works to select of promising individuals with high quality from many seedlings [12,17].

## 3. Suggestions from the Citrus Genealogy for Breeding

### 3.1. Genomic Breeding Aims to Avoid the Breeding Trilemma

In citrus breeding, the longest part of the breeding period consists of the long juvenile period until first blooming (6–7 years) and the period required for trait evaluation in the field. GAB is an emerging technology to improve breeding efficiency by rapidly selecting promising individuals of high quality [17,24]. This approach aims to avoid the breeding trilemma, but selection by MAS and GS alone is not sufficient to shorten the time required for breeding. 'Fast breeding' that intends to develop unique, unconventional cultivars in the shortest possible time is another constraint to the breeding trilemma [13]. Minimizing the time from crossing to the selection of an elite hybrid in each generation would be effective in achieving fast breeding. However, most citrus cultivars have long juvenile periods, and alternate bearing that hampers the trait evaluation in the short term is severe in young citrus trees. It extends the period for repeating the trait evaluation to maintain accuracy, making the whole breeding process long. The citrus breeding program in NARO evaluates over 20 traits. The current prediction accuracy in citrus by GS is up to 80% for particular traits, but other traits do not reach this level [30]. That means traits that MAS or GS cannot select or are less predictable need to be evaluated in the field, along with confirming the traits selected by MAS or GS. Therefore, even if using GAB, a drastic reduction in the breeding period cannot be expected because both MAS and GS are the selection method, not the methods that shorten the juvenile period. Furthermore, considering the period required for a single generation (over 20 years), achieving a breeding process that had required several generations by the conventional methods within a single generation will drastically shorten the time required for developing a cultivar.

The conventional breeding program often uses existing superior cultivars as parents. Although this approach ensures that breeders develop a new hybrid of high quality in a short period, crosses between similar cultivars lower the novelty of the hybrid breed. Using cultivars not used as breeding parents to cross may contribute to enlarging the diversity of hybrids and selecting a novel individual. While providing these unfamiliar cultivars for breeding is expected to contribute to developing individuals with novel traits, the multiple unwanted traits of the parent may decrease the ratio of individuals that exceed the selection threshold. Although few studies have examined the inheritance of quantitative fruit traits in citrus, Combrink et al. showed the continuous distribution of the height and width of fruit in six offspring populations developed from different cultivars, using Kiyomi as the common female plant [31]. They used six cultivars whose fruit sizes are smaller than Kiyomi for the male plants. The study showed that the medians of fruit widths and height of the six populations were close to that of another parent that set fruit smaller than Kiyomi. Their results strongly suggested that more seedlings could be required to select individuals of larger fruit sizes than Kiyomi when crossing Kiyomi with a small fruit cultivar. In contrast, these parental cultivars' fruit shape indexes (ratio of fruit width per fruit height) showed no considerable differences and a narrow distribution. Although medians of the fruit shape indexes of these populations were close to those of parental cultivars, a few individuals showed substantial differences from their parents for the fruit shape index [31]. They also revealed that rind color factors ($L*$ and $a*$) of the same six populations of Kiyomi showed similar distributions in their offspring populations as observed in fruit size in another study [32].

Fruit size is one of the primary traits for selection in the NARO citrus breeding program that uses Kiyomi to cross frequently, as demonstrated in Figure 1. Many cultivars that

have not been used for breeding set small fruits in the program. Thus, as demonstrated by Combrink et al. [31], the average fruit size of the population is smaller than Kiyomi and is an obstacle when using a small fruit cultivar for crossing to expand the diversity. These results were examined for five fruit traits in populations of Kiyomi as the common parent, and great care must be taken to expand the finding to all crosses and fruit traits. However, these results strongly suggest that it is challenging in conventional breeding to develop new hybrid cultivars that exceed the threshold on the broad traits in a single generation because using non-elite cultivars would suppress the occurrence of promising individuals in a population.

For example, NARO recently released the 'Aurastar' [33], 'Nou No. 7', and 'Nou No. 8' cultivars [34]. These are second-generation trifoliate oranges (TF) obtained by crossing Hassaku × TF 'Flying dragon' followed by crossing with Banpeiyu pummelo or Kiyomi to introduce resistance to the citrus tristeza virus of TF into the commercial cultivars [33,34]. Although these have improved fruit traits compared to TF, they retain the unpleasant odor of TF and are inedible. Thus, they require further introgression of genomes of edible citrus cultivars. This is a typical example of the breeding trilemma. Serra et al. proposed an approach (Marker-Assisted Introgression; MAI) to shorten the generation required for the introgression of a distant cross with MAS in peach [35]. Despite the use of the MAI approach, it requires several generations and avoiding a lengthy breeding period is difficult in citrus. Therefore, a different approach is needed to resolve the trilemma.

### 3.2. Lessons from the Revealed Genealogy of Citrus Cultivars

Understanding how existing indigenous citrus cultivars have been selected would provide clues to improve breeding efficiency and resolve the trilemma. A recent study revealed that modern citrus species emerged from iterated hybridization of ancestral citrus cultivars between China and India approximately 9 million years ago [36,37]. Although the process by which current indigenous citrus cultivars were established had long been unknown, Shimizu et al. revealed the pedigree of over 60 of these cultivars [38]. Tanaka classified more than 160 indigenous citrus 'species' [39,40]. However, these results confirmed that most of these 'species' are natural hybrids and thus cannot be classified as individual species [38].

The study of citrus genealogy provided important suggestions: (1) Repeated crossing of several key cultivars, such as Kishu, Kunenbo, Yuzu, and pummelo, has led to the current diverse group of cultivars within a few generations. This indicates that the genetic diversity of those indigenous cultivars has decreased, and Shimizu et al. [38] confirmed this using SSR marker analysis. (2) Fruit traits such as size, shape, color, rind thickness, SSC, acidity, and peelability had changed significantly in just one generation. This suggests that repeated generations are not always essential to produce an elite hybrid. (3) The elucidated genealogies indicate that cultivars not previously used in breeding were the parents of several unique cultivars. Using these unfamiliar exotic cultivars as breeding parents would be helpful in developing new cultivars with novel traits.

### 3.3. Hybrids Selected from Kishu and Kunenbo

Kishu (*C. kinokuni* hort. ex Tanaka) is an ancient cultivar that has been known since at least 740 A.D. in Japan [41]. Shimizu et al. demonstrated that Kishu served as the direct parent of 12 cultivars (Kunenbo, Satsuma, Yatsushiro, Naruto, Oukan, Natsudaidai, Nidonari mikan, Fukure mikan, Suruga Yuko, Sokitsu, Sanbokan, and Andokan) and as the indirect parent of 24 cultivars (Jabara, Mochiyu, Kabosu, Henka mikan, Kizu, Konkoji, Hyoukan, Hassaku, Asahikan, Kaikoukan, Yamabuki, Iyo, USSR tangelo, Shunkokan, Kawabata, Unzoki, Kabuchi, Keraji, Yuge hyoukan, Ujutkitsu, Kawachi bankan, Sanbokan, Andokan, and Yuukunibu) through Naruto (*C. medioglobosa* hort. ex Tanaka) or Kunenbo (*C. nobilis* Lour. var. kunep Tanaka) [38]. Kunenbo is an offspring of Kishu, crossed with an unidentified seed parent [38]. This was confirmed using whole-genome sequencing analysis [42].

Comparing the parental Kishu and Kunenbo with their offspring cultivars shows the changes in fruit size, rind color, and rind smoothness between them (Figure 3). Fruit cross-sections also show the changes in flesh color, rind thickness, and the number of seeds (Figure S1). Although not apparent from the images, variation is also observed among these cultivars in many traits, such as sugar and acid content, coloring time and aroma, and flesh quality. These trait variations do not differ from those found in the population of a single cross in citrus. The probability of obtaining an elite candidate with superior traits, such as Satsuma, from a single cross between Kishu and Kunenbo is supposed to be very low. Therefore, with the current systematic breeding program, the possibility of selecting an individual with traits comparable to those of Satsuma is insignificant.

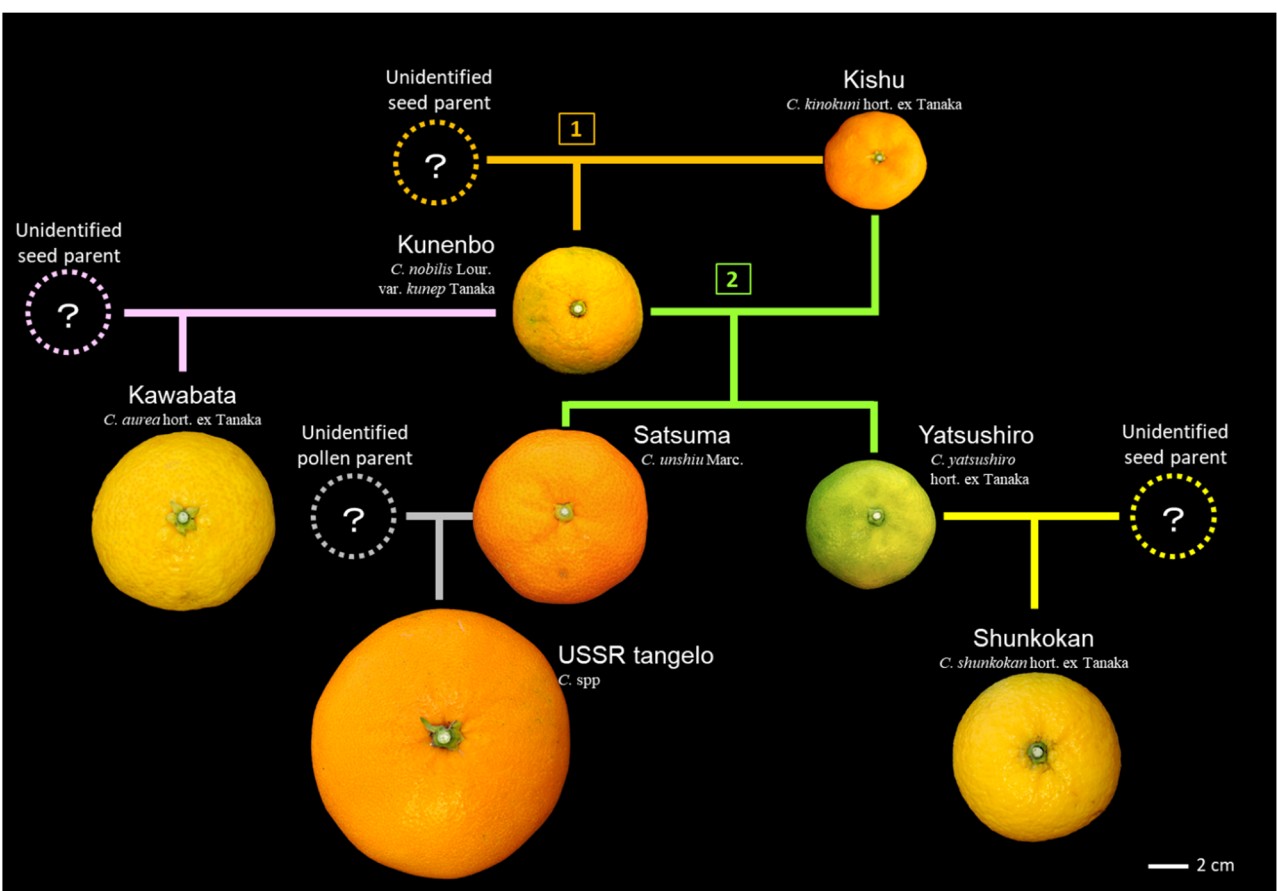

**Figure 3.** Pedigree of Kishu (*C. kinokuni* hort. ex Tanaka) and Kunenbo (*C. nobilis* Lour. var. kunep Tanaka). A question mark indicates an unidentified parent. The numbers in squares indicate the first cross of Kishu to produce Kunenbo (1), then the second cross of Kishu with Kunenbo to produce Satsuma and Yatsushiro (2).

Satsuma was discovered in Japan around 1648 [43]. One factor that enabled the selection of Satsuma from a cross between Kishu and Kunenbo when no systematic breeding system was available was adjacent cultivation of these that facilitate natural hybridization [41]. There were many such areas in Japan with seeds obtained and discarded every year. Satsuma would be selected over several decades from the seedlings germinated from these discarded seeds. This indicates that it is possible to select a high-quality elite hybrid within a single generation by taking sufficient time for selection, even without conducting modern systematic breeding.

### 3.4. Hybrids Selected from Yuzu

Yuzu is another ancient cultivar already common in Japan by 717 A.D. [41]. Shimizu et al. demonstrated that Yuzu served as a parent of 10 cultivars (Sudachi, Kourai tachibana, Ichang lemon, Jabon, Hanayu, Jabara, Mochiyu, Kabosu, Henka mikan, and Kizu), and five of those cultivars were hybrids with Kunenbo [38]. This suggests that Yuzu and Kunenbo were grown adjacent in the same orchard for an extended period. These offspring of Yuzu are distinctive acid citrus cultivars. As with Kishu and its offspring (Figure 3 and Figure S1), differences in fruit size, shape, rind, and flesh color were observed between Yuzu and its offspring (Figure 4 and Figure S2). In addition, Yuzu-like rind puffing is observed in Kourai tachibana, Ichang lemon, Kabosu, Sudachi, and Mochiyu. Of those, Sudachi and Kabosu retain a Yuzu-like aroma [39]. The aroma of Sudachi is like that of Yuzu, but stronger and fresher. However, Kabosu has a milder, sweeter aroma than Yuzu. These were previously considered Yuzu relatives [39,44]; however, they are now known to be chance seedlings of natural hybrids of Yuzu [38]. Although systematic breeding focusing on flavor has so far proven difficult, it is empirically known that offspring with fragrances similar to their parent, such as Yuzu and sweet orange, often emerge [12,44]. The revealed parentages among Yuzu and its offspring that harbor a Yuzu-like aroma suggest it is possible to breed new cultivars that inherit the characteristic flavor from a parental cultivar with a distinctive fragrance like Yuzu within a single generation.

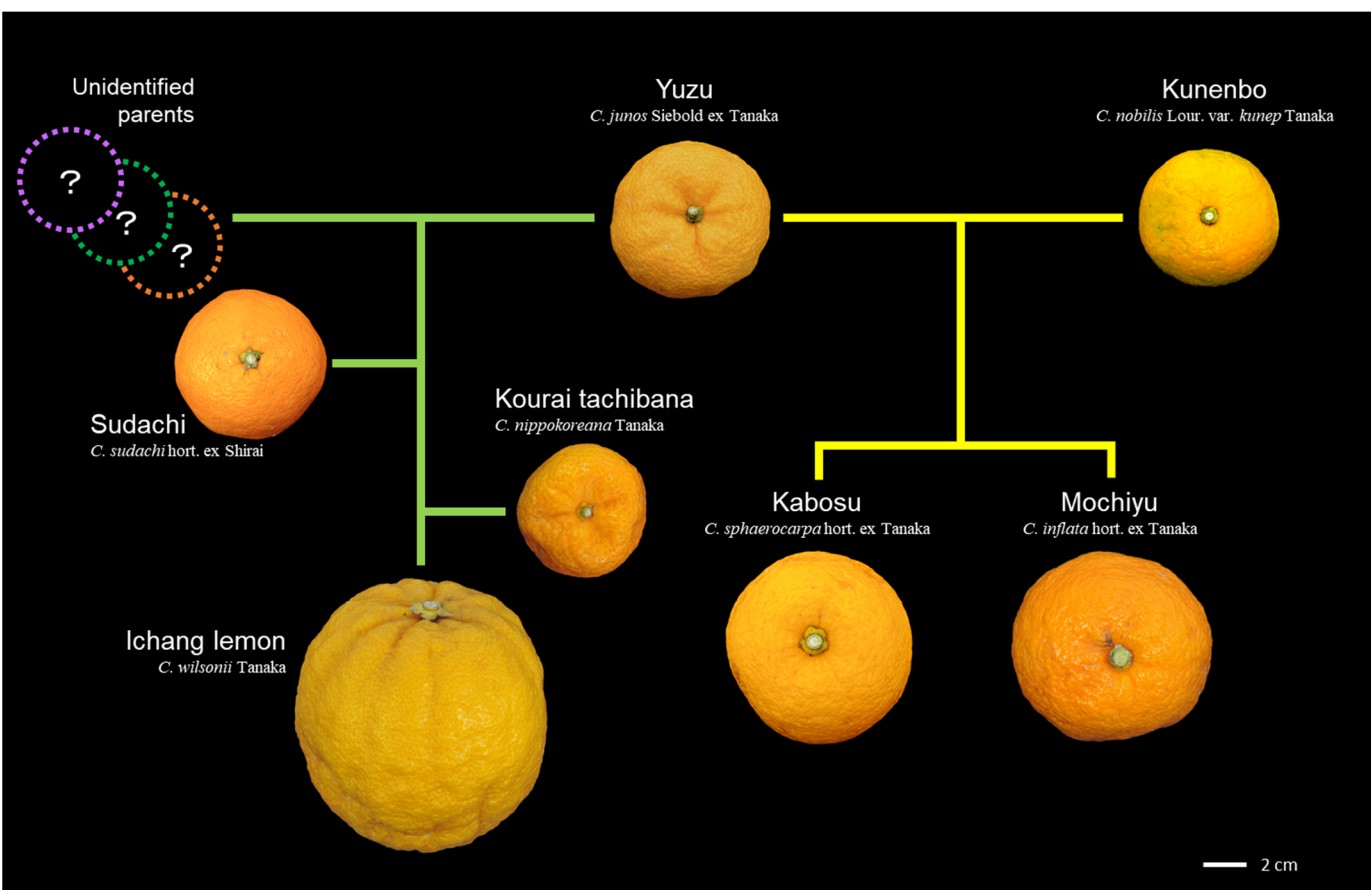

**Figure 4.** Pedigree of Yuzu (*C. junos* Siebold ex Tanaka) and Kunenbo.

### 3.5. Hybrid Selected from Kunenbo

Kunenbo (*C. nobilis* Lour. var. kunep Tanaka) is another ancient cultivar that may have been established in Japan in the 8th century [41]. Shimizu et al. identified four *C. nobilis* genotypes (Kunenbo A, Kunenbo B, Twukkuni, and King mandarin) [38]. Among those *C. nobilis* strains, Kunenbo A (hereafter denoted as Kunenbo) is the major genotype

that served as the direct parent of 16 cultivars (Satsuma, Yatsushiro, Kawabata, Unzoki, Kabuchi, Keraji, Jabara, Mochiyu, Kabosu, Henka mikan, Kizu, Konkoji, Hyoukan, Hassaku, Asahikan, and Kaikoukan) and as the indirect parent of seven cultivars (Yamabuki, Iyo, USSR tangelo, Shunkokan, Sanbokan, Andokan, and Yuukunibu) [38]. Of these, five offspring cultivars (Hyoukan, Kinkoji, Kaikoukan, Asahikan, and Hassaku) have a pummelo genotype that indicates that they are hybrids of pummelo with Kunenbo (Figure 5). These five cultivars show pummelo-like characteristics such as large fruit size, thick rind, and a pummelo-like flesh texture. The rinds of Hyoukan, Kinkoji, and Asahikan are orange, similar to mandarin or Kunenbo, and all bare orange flesh except for Kaikoukan (Figure 5 and Figure S3).

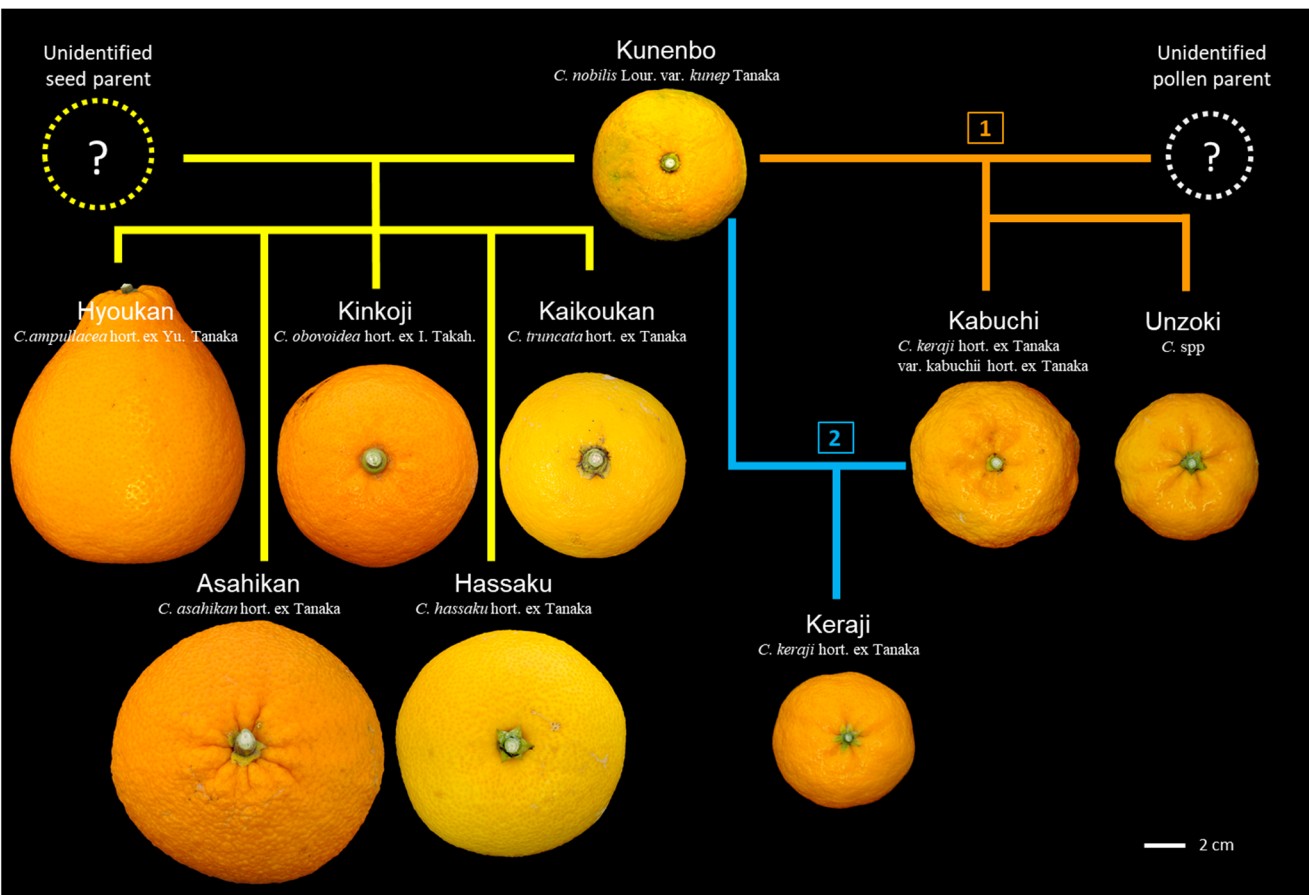

**Figure 5.** Pedigree of Kunenbo (*C. nobilis* Lour. var. kunep Tanaka). The one in the square indicates the first cross of Kishu, and the two in the square indicates the second cross of Kishu.

Three cultivars (Kabuchi, Unzoki, and Keraji) that harbor the same cytotype as Kunenbo are small and show thin, orange-colored rind and flesh, and rind puffing typical of a mandarin (Figure 5 and Figure S3). These differences found in fruit traits that correlate with cytotype strongly suggest that the cytotype of the cultivar used as the female parent also affects the fruit traits of the hybrid. Hyoukan is a hybrid from a cross between an unidentified pummelo and Kunenbo; however, the unique ovoid shape of the fruit of Hyoukan is not reminiscent of its parent, Kunenbo. The change in the fruit shape of Hyoukan again indicates that a single cross is enough to result in significant changes in fruit traits, as observed in Kishu.

### 3.6. Hybrids Selected from Kaikoukan

Kaikoukan (*C. truncata* hort. ex Tanaka) is a forgotten cultivar whose importance has never been well understood. However, it served as the seed parent of four unique cultivars (Yamabuki, Iyo, Sanbokan, and Andokan) and indirectly served as the parent for Yuukunibu (Figure 6). Kaikoukan was rarely cultivated, but its offspring, Sanbokan and Andokan, were once widely grown in several regions, and Iyo remains popular to this day. Even among the parent and offspring cultivars, many fruit traits, such as fruit size, rind and flesh color, rind thickness, and flesh texture, have changed significantly within a single generation. This suggests that Kaikoukan, which has not been used in breeding before, has significant potential as a breeding parent. Furthermore, drastic changes in fruit shape were observed in Sanbokan, Andokan, and Yuukunibu compared with Kaikoukan, such as a unique 'neck' (Sanbokan) or a flattened oval shape (Andokan and Yuukunibu) (Figure 6). These changes in fruit shape are reminiscent of Hyoukan (Figure 5). Comparing the fruit traits of the offspring of Kaikoukan (except for Yamabuki, whose other parent is unknown), these cultivars bear the same orange flesh color as their parent, in contrast to the yellow-fleshed Kaikoukan (Figure S3). However, the rind of these four cultivars is as thick as that of Kaikoukan, and none have a thin rind similar to that of the parental mandarins Kishu or Dancy as observed in Kunenbo and its offspring. These observations suggest each trait is independently inherited in Kaikoukan and its offspring, with the rind being affected by the cytoplasm and the flesh color by the pollen parent, as implied by their cytoplasm types [38]. Kaikoukan could also serve as a breeding parent to deliver a highly distinct fruit shape to its offspring if its shape is also considered valid as a variety feature.

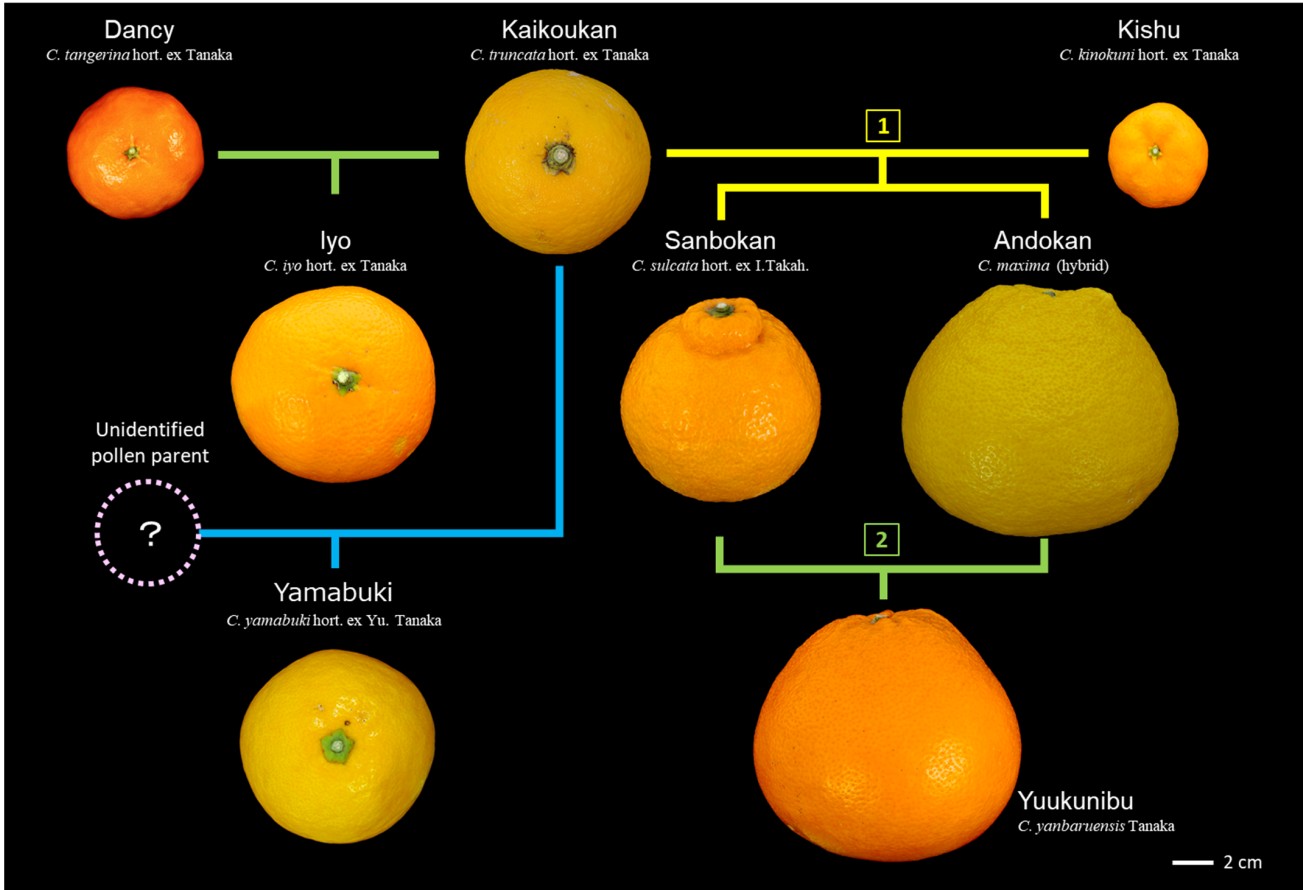

**Figure 6.** Pedigree of Kaikoukan (*C. truncata* hort. ex Tanaka) and Kishu with Dancy. The one in the square indicates the first cross to produce Sanbokan and Andokan, and the two in the square indicate their crossing.

### 3.7. Hybrids Selected from Tachibana

Tachibana (*C. tachibana* (Makino) Tanaka) is considered to have been introduced to Japan in 70 A.D., and is regarded as the oldest citrus variety in Japan [41]. Early studies attempting to classify tachibana implicitly assumed that it comprised a single genotype [39,40,45]. Later, based on leaf isozyme analysis, Hirai et al. reported the existence of four strains in Japan [46]. SSR marker analysis of broad citrus genetic resources by Shimizu et al. revealed three genetically different tachibana strains in Japan and four offspring varieties (Hanayu, Hyuganatsu, Girimikan, and Oogonkan) [38]. Although the parents of three offspring (Hyuganatsu, Girimikan, and Oogonkan) are not known, those unidentified parents served as the seed parent [38]. Shimizu et al. reported that tachibana A is polyembryonic, but no hybrid was found among seedlings when the genotype of seedlings was detected using SSR marker analysis. This suggests that it could be difficult to obtain hybrids from tachibana seeds [47]. Differences between three tachibana strains and their offspring were observed in many traits, including fruit size, rind and flesh color, and rind thickness (Figure 7 and Figure S5). These differences are more evident than those observed between Kishu, Kunembo, Kaikoukan, and their offspring (Figures 3–7), which may explain why these four varieties have not been recognized as tachibana hybrids. Tanaka classified Hanayu as *C. hanayu* Sieb. ex Shirai, and had been thought to be a Yuzu-related variety [39]; however, it was later discovered to be a natural hybrid of Yuzu and tachibana A. Despite the differences in fruit traits between them, the flavor of Hanayu is like that of Yuzu but not tachibana; therefore, it has been used as a replacement for Yuzu.

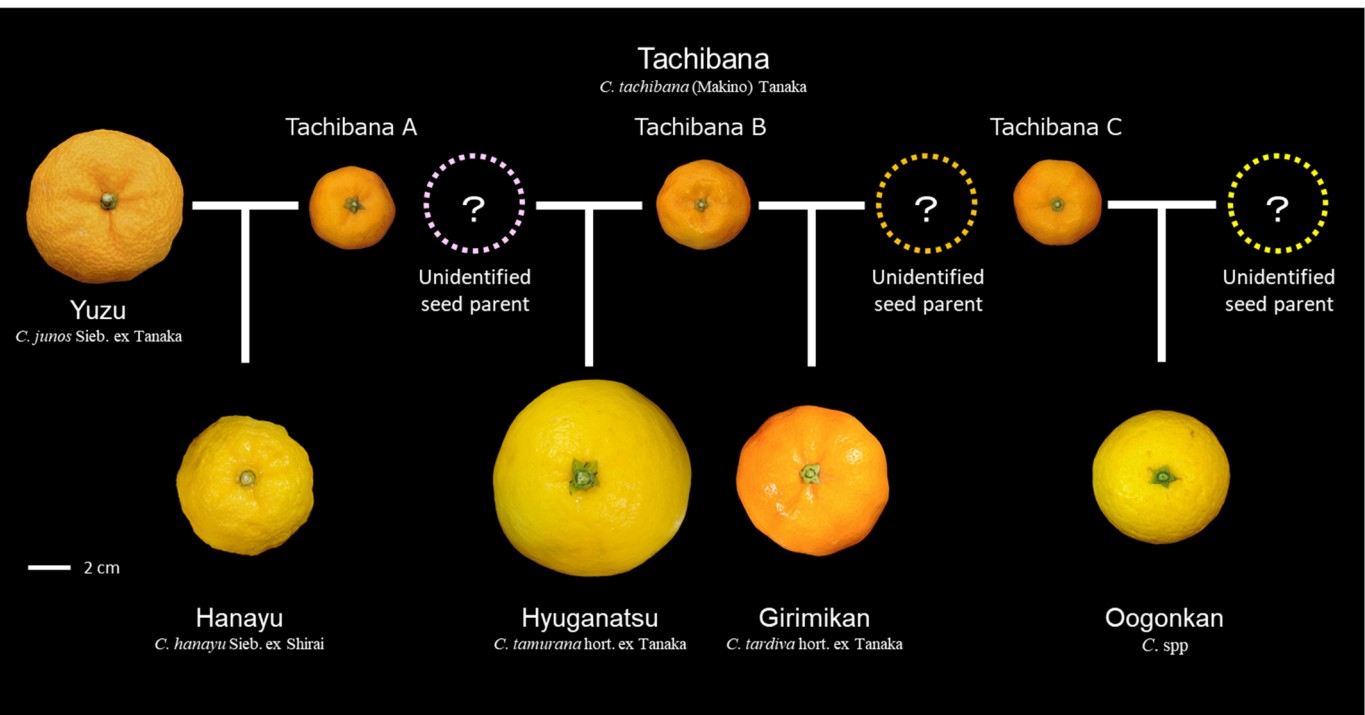

**Figure 7.** Pedigree of three tachibana strains.

These tachibana strains have never been used as breeding parents owing to the small, highly acidic, seedy, and rind puffing traits of the fruit. However, the offspring of tachibana have a unique aroma and flavor not found in mandarins, and the fruit is of an appropriate size. These four offspring varieties were all selected from a single cross of tachibana, which implies that distinguishing varieties can be selected within a single generation from tachibana crosses.

## 4. Integration of Genomic Breeding with Genealogy to Avoid the Breeding Trilemma

Genomics-assisted breeding using MAS and GS is the key to solving the breeding trilemma by improving the efficiency of selecting elite hybrids with premium quality, but the challenge remains to balance the expansion of diversity with fast breeding. From the elucidated genealogy, cultivars with high quality and uniqueness, such as Satsuma, Kabosu, Sudachi, Hassaku, and Hyuganatsu, were selected within a single cross. Furthermore, as discussed in Section 3.3, the revealed genealogy indicates that evaluating many seedlings over a sufficient period enables the selection of a high-quality elite hybrid from a single cross, even though it is unlikely in a short span of time. In citrus breeding, increasing diversity is a conflicting goal to improve quality. As has been observed in several families, especially in Kaikoukan and tachibana with their offspring, even a single generation of crosses is sufficient to select a high-quality elite hybrid that significantly differs in its fruit trait from the parent. Now, genome-wide prediction techniques by GS have become available for improving the breeding efficiency in citrus by selecting many seedlings in a short period. Combining GAB techniques to select thousands to tens of thousands of seedlings makes it possible to breed new cultivars with high quality and distinctive traits in a single generation, and the breeding trilemma can be avoided. Using the genomic breeding approach contributes to increasing the virtual breeding size, enhancing the chance to select the most ideal seedling while reducing the breeding cost.

## 5. Future Perspective

A GAB approach that considers the revealed genealogy of citrus will enable different approaches not previously possible in conventional breeding—rebreeding, complementary breeding, and mimic breeding.

Rebreeding aims to reproduce the existing cultivar through hybridization. This approach selects a hybrid with a nearly identical genotype to the existing cultivar by large-scale genotyping of many seedlings. The rebreeding approach intends to provide a new strategy for mutation breeding by selecting new hybrids with traits that are equal or superior to those of the original cultivar. Eduardo et al. proposed a similar approach, called resynthesis, to select elite genotypes using marker-based partial reconstruction [48]. They demonstrated the potential of this approach in peach by selecting seven individuals 76–94% identical to the reference from the 416 F2 population. The resynthesis approach, which selects genotypes close to the target cultivar from multiple generations of selection, can be an effective method for peaches with a short juvenile period. However, it is not feasible to select individuals close to the target genotype by repeating the cross because of the higher heterozygosity and longer generation time of citrus than peach. Therefore, it is desirable to complete the selection in a single generation; but evaluating a large set of seedlings is mandatory. Selection of the seedlings before grafting by genome-wide genotyping analysis is suited for such a purpose.

Complementary breeding is similar to rebreeding, but incorporates the selection of a particular trait with GWAS and GS to further improve the existing cultivar. Although many sports or nucellar seedlings have been selected in most cultivars to improve a minor trait in citrus, the probability of the occurrence of mutations is low, and individuals with the desired mutation are not always obtained. Complementary breeding initially selects the traits to be improved in a set of seedlings from the same cross of the target cultivar by MAS with DNA markers developed from GWAS. Then, GS selects the candidate individuals that resemble their traits to the target cultivar. The advantage of this approach is that a hybrid with improved target traits can be obtained in a short period. In peach, MAI which is a similar approach, actively uses MAS reported for the introgression of almond genes into peach [35]. However, these approaches require multiple generations and are challenging to implement in citrus because they drastically extend the breeding period. Selection in the shortest possible generation for citrus cultivars with a more extended juvenile period than peaches effectively shortens the breeding period.

Another approach, mimic breeding, focuses on selecting new hybrids by MAS and GS by mimicking the revealed parental combination. The procedure of mimic breeding is similar to that of complementary breeding, but it does not aim to select individuals close to a particular existing cultivar for their traits but the best candidate individuals that fit the breeding target. By referring to a known cross combination, it enables the development of a similar but unique new hybrid with premium quality. Integrating the MAS or GAB technique during mimic breeding facilitates the improvement of a particular trait of the target cultivar in a short period. These new breeding approaches that integrate GAB and the citrus genealogy, collectively termed "Citrus Breeding 2.0" [13], are underway in NARO. Though it has not completed developing cultivar, ongoing evaluation found this approach using MAS/GS works to predict promising candidates. Another report will be submitted after developing elite cultivars and verifying the whole process.

The cost is another essential issue in conducting GAB. Although the genome-wide genotyping cost has been declining, the expense of genotyping all seedlings is too high to be ignored. However, the two-step selection of seedlings before grafting will contribute to reducing the entire cost of breeding. In the two-step selection, MAS for a large set of seedlings using several DNA markers preliminary selects the primary candidate seedlings and then eliminates most of the hopeless individuals. In this first selection, SSR or SNP markers developed from GWAS are suitable for low-cost selection of a large population. In the next second selection, genome-wide genotypes of the few remaining promising individuals are obtained and then provide them with GS prediction of fruit traits to select elite individuals. The total cost of citrus breeding to maintain and evaluate many individuals for an extended period is much higher than annual crops. Therefore, conducting GAB in a two-step manner will suppress the total cost compared to the cost of the phenotypic selection of many seedlings.

The success of these approaches relies on the accuracy of selection, and the variation of selectable traits by MAS or GS. Cross combination is another factor that affects the probability of selecting a target individual. Minamikawa et al. recently revealed in citrus that increasing the number of samples improved the prediction accuracy in GS and the number of significant SNPs [30]. Zhang et al. reported a similar positive correlation of marker density with prediction accuracy in maize, cattle, and pig populations, despite statistical methods for prediction that also affect accuracy [49]. Such efforts appear to conflict with improving breeding efficiency. However, enriching the genomic diversity of the plants used for modeling is anticipated to improve the prediction accuracy of the traits on a broad cross combination. Because the probability of obtaining elite hybrids is supposed to decrease when using exotic cultivars for the cross, predicting the expected probability of promising ones from a cross makes it possible to estimate the population size to be selected in MAS in advance. The preliminary assessment of cross combination hopes to enable simulation-based breeding and contributes to conducting breeding in a planned manner. Simulation-based breeding will also enable focusing on the cross expected to have the highest probability of producing promising hybrids based on a simulation.

## 6. Conclusions

Seedling selection by GAB that integrates MAS and trait prediction by GS is anticipated to enhance citrus breeding. Although these new techniques are anticipated to improve the whole breeding process, the question of how to develop new premium quality cultivars that satisfy broad consumer demands in a short period remained. The study of the genealogy of indigenous citrus cultivars proposed an approach to avoid the breeding trilemma that has been a constraint of conventional breeding. These efforts will help to avoid the breeding trilemma and breeding cultivars of high quality and diversity with minimum effort and duration. This new approach, "Citrus breeding 2.0," is anticipated to enable the development of new cultivars with novelty and high quality in a short period through systematic breeding programs comprising rebreeding, complementary breeding, and mimic breeding.

**Supplementary Materials:** The following supporting information can be downloaded at: https://www.mdpi.com/article/10.3390/horticulturae8060559/s1, Figure S1: Fruit cross-section of Kishu and Kunenbo with their offspring cultivars, Figure S2: Fruit cross-section of Yuzu and Kunenbo with their offspring cultivars, Figure S3: Fruit cross-section of Kunenbo with their offspring cultivars, Figure S4: Fruit cross-section of Kaikoukan with their offspring cultivars, Figure S5: Fruit cross-section of three tachibana strains with their offspring cultivars.

**Funding:** This research was funded by JSPS KAKENHI C 18K05634, KAKENHI B 20H02980, Next-Generation Breeding project (NGB2009) from the Japanese Ministry of Agriculture, Forestry and Fisheries for NGB2009, and Cabinet Office, Government of Japan, Cross-ministerial Strategic Innovation Promotion Program (SIP), "Technologies for Smart Bio-industry and Agriculture" (funding agency: Bio-oriented Technology Research Advancement Institution, NARO) for DDB2001.

**Acknowledgments:** Authors acknowledge Keisuke Nonaka, Akira Kitajima, Mai Minamikawa, Hiroyoshi Iwata, and Eli Kaminuma for their helpful discussions.

**Conflicts of Interest:** The author declares no conflict of interest.

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
