# Peer review of "Breeding New Premium Quality Cultivars by Citrus Breeding 2.0 in Japan: An Integrative Approach Suggested by Genealogy"

_horticulturae, doi:10.3390/horticulturae8060559_

Round 1

Reviewer 1 Report

Excellent review, a lot of informations on the origin of Japanese citrus varieties.

The proposed new strategy for making faster and  more efficient  citrus genetic improvement  is very interesting.

Author Response

Dear Reviewer 1,

Thank you for taking the time to review my manuscript, and I sincerely appreciate your kind comment. I am glad to hear your positive comments, and hoping you may found this short review is useful. 

I have revised the manuscript as to the suggestions from other reviewers. I would appreciate it if you could confirm the revised manuscript to make it easier to understand and sound.

Sincerely, 
Tokurou Shimizu

Reviewer 2 Report

The manuscript proposes a strategy for addressing major challenges in conventional citrus breeding by integrating genome-wide prediction and genealogy. Historical outcomes of the NARO breeding program are specifically used to exemplify principles. Pedigrees of certain cultivars are investigated to support claims that large genetic gains can be made in few generations. Specific applications of “rebreeding”, “complementary breeding”, and “mimic breeding” are briefly mentioned.

The manuscript is written in clear and usually logical manner, with interesting and high-quality figures. Appropriate background is provided in the Introduction and the need for a new paradigm in citrus breeding is reasoned. However, there are numerous lapses in logic and lack of adequate referencing in key places that will require deeper argumentation and stronger scientific support (including much more from relevant crops other than citrus). Review of other citrus breeding programs for which the 2.0 strategy would be appropriate is needed. Even after reviewing evidence for the feasibility of the 2.0 strategy (which in any case appear to be limited), the overall thesis of this manuscript should be scaled back to being a proposition, a hypothesis, rather than an emphatic “this will work!”. This manuscript masquerades as a review but in its current form is not a review.

As pleasant as they are to look at, the figures showing the cross sections should be moved to supplementary material or else combine the paired figures to show side-by-side for each variety both its outer view and its cross section.

Lines 38-39: I disagree with the implication that the success of crossbreeding relies entirely on either breeder experience or chance. What about just making sound decisions based on accurate information? Having and using information does not absolutely require experience.

Line 41: “Crossbreeding” is not somatic (asexual) hybridization, but rather it is gametic/sexual hybridization.

Line 58: Should be “Fruit breeding programs” and there should be background provided of other citrus breeding programs. It is unclear why suddenly the NARO program is described without broader context. The lack of context leads to blurring throughout the rest of the manuscript because it is unclear whether the principles or strategies described are referring to the NARO program or any citrus breeding program. For example, line 206 “The conventional breeding program” – NARO or any program? The heading above the paragraphs describing the NARO program should then mention that program – currently the heading on line 57 is misleading.

Line 62, 73, 78, 117, and elsewhere: Not appropriate to refer to “we”/”our” – who is that? Can’t be the team of every author on this manuscript because there is only one author. Avoid such pronouns. The “we” used in line 132 is even more problematic, as it seems to be trying to refer to all citrus breeders or even all conventional breeders of any crop. Also problematic on line 237.

Lines 64-73, 87-91, and elsewhere should be past tense.

Line 72, 86, Table 1: “Brix” (and actually degrees Brix) is a unit of the measurement called SSC (just like “grams”/’g” is a unit of the measurement of fruit size). Therefore, “high Brix content” is incorrect terminology.

Line 89: The adjective “good” is vague, non-scientific.

Line 113: Should be “alternate”, not “alternative”

Lines 126-132: Without any citations of the statements made, it is unclear whether this part is referring to the NARO program or to citrus breeding in general. This paragraph needs referencing to be includable in this review. In fact, this paragraph is the crux of the argument of the manuscript, as it is where is provided the justification for a paradigm shift in citrus breeding. But without any references, it is entirely a “trust me” argument, which is not acceptable for a scientific review paper.

Line 142: What is “genomic breeding”? Used in the heading but not yet defined nor defined in this section. The word “demand” is probably not appropriate here – something like “opportunity” would fit better.

Line 148-150: Meaning of this statement is unclear. The two parts of the sentence make sense, but it is unclear how they logically connect together.

Line 150-152: This argument needs referencing.

Line 153: “will dramatically increase the virtual breeding scale” – a strong statement that carries little weight without definition of the terms used here, clearer argumentation, and citation of appropriate studies backing it up.

Line 171-188: GWAS and GS are first appropriately defined as two separate things but then they are intermingled from sentence to sentence. The conclusions made on lines 186-189 are therefore not adequately supported. The relevance of each of these two things, separately, to citrus breeding needs to be adequately explained. For example, GS is stated to avoid the concern of segregation of specific trait-associated loci, yet GWAS is stated to detect trait-associated loci. Why not just use GS and avoid GWAS? GWAS is useful towards “conventional” MAS, but not GS. Line 195 further highlights the confusion: “selection by GWAS” does not make sense. In fact, it looks to me that GWAS should not be part of the “2.0” strategy of this manuscript. Lines 432 and 436-7 incorrectly connect the two together immediately as if they are the same thing or as if they easily click in together.

Line 181: “linked to” should be “associated with”. The term “linked”, i.e., regarding genetic linkage, refers to connections between loci – such as marker loci and trait loci. A locus cannot be “linked” to a trait such as fruit size or even a phenotype such as large fruit size – but it can be associated with such.

Line 191: Reword this heading – the grammar has several problems.

Line 196-197: How can fast breeding be a constraint? The constraint is the long juvenile phase, not the proposed solution to overcome it.

Line 200: What is the problem with alternate bearing for GAB? Why not use GAB to select against hybrids predicted to have that undesirable phenotype?

Lines 206-208: I don’t understand the unification made between “existing superior varieties” and “similar varieties”. Just because two varieties are superior does not mean they are genetically similar.

Lines 206-217: Just as for lines 126-132, this paragraph should not be included without more referencing. Every statement of fact should be referenced, while statements of reason should build on the facts in a very logical way that avoids any hint of “trust me, reader”.

Line 272-274: The meaning of “chance” here is not clear. Whether there is a high probability of achieving a superior hybrid in a single cross or a low probability, it’s still highly dependent on “chance” because the specific outcomes depend on uncontrolled recombination, segregation, etc. “Chance” (just like “probability”) is a neutral term that requires a qualifier (high, low, etc.).

Line 304: “it is known empirically” – by whom? Reference?

Line 360: Unclear why “cytoplasm” is referred to. Should be “maternal parent”. Also, “affected” is not the right word at all, and indicates lack of understanding of inheritance principles. The alleles for any phenotype are inherited from both parents (except in unusual circumstances), so both parents “affect” any trait. Perhaps in this sentence it is trying to refer to the presence and inheritance of dominant-effect alleles.

Lines 421-430: The argument of this paragraph does not make sense. As stated, rebreeding requires multiple generations. Yet the lead-up to this section is about how citrus breeding does not need multiple generations and especially that would be greatly hampered by and instead requires a solution that bypasses long juvenility. The solution seems to be complementary breeding, in the next paragraph. But how can “a hybrid with improved target traits can be obtained in a short period”? The only evidence provided is in peach-almond that requires multiple generations via MAI.

Lines 432 and 436-7: As mentioned previously, the inclusion of GWAS here does not make sense. GWAS is not about selection.

Line 437: As the manuscript previously describes, GS requires genotyping with genome-wide markers. How is this to be practically and affordably done with “numerous seedlings”?

The difference between rebreeding and complementary breeding is not adequately explained. The supposed differences mentioned on lines 431-2 (“but…”) are already components mentioned in rebreeding.

Lines 444-448: This paragraph does not adequately describe what mimic breeding is. Nor does it provide any evidence that this is a feasible strategy.

Line 451: What does MAS have to do with the success of the approaches? Earlier in the manuscript, MAS was specifically noted as being “conventional” and needing replacement by a new strategy. And what follows in this paragraph is not about MAS anyway, but rather GS.

Lines 456-464: These are all just “trust me” statements, and thus carry little weight.

Lines 468-475: This Conclusion is much too over-reaching. These strong statements are not supported by the rest of the manuscript. There is no evidence provided that the proposed “GAB” strategy has been or will be effective.

Author Response

Dear Reviewer 2,

Thank you for taking the time to carefully review my manuscript, and I sincerely appreciate your kind comment and suggestions. These comments and suggestions were quite helpful in improving the quality of this manuscript and its readability. I revised the manuscript, and anticipated those changes follow the comments and suggestions. Please confirm the following reply for each suggestion and comment.

# The manuscript is written in clear and usually logical manner, with interesting and high-quality figures. Appropriate background is provided in the Introduction and the need for a new paradigm in citrus breeding is reasoned. However, there are numerous lapses in logic and lack of adequate referencing in key places that will require deeper argumentation and stronger scientific support (including much more from relevant crops other than citrus). Review of other citrus breeding programs for which the 2.0 strategy would be appropriate is needed. Even after reviewing evidence for the feasibility of the 2.0 strategy (which in any case appear to be limited), the overall thesis of this manuscript should be scaled back to being a proposition, a hypothesis, rather than an emphatic “this will work!”. This manuscript masquerades as a review but in its current form is not a review.
Thank you for the suggestion. I revised the manuscript thoroughly, included references to support my points, and added new paragraphs to make up not fully explained sentences (sentences in red).

# As pleasant as they are to look at, the figures showing the cross sections should be moved to supplementary material or else combine the paired figures to show side-by-side for each variety both its outer view and its cross section.
Thank you for the suggestion. I tried to combine each pair of whole fruit and cross-section, but that made the figure complicated to understand. Since fruit cross-sections are also repeatedly mentioned in the text, I worry that moving them to the supplement would hamper the reader's understanding. Thus, I decided not to revise these figures. I hope you may agree this. I understand Horticulture Journal has no page limit, but I may reconsider if the editor-in-chief asks me to consider it again for technical reasons.

# Lines 38-39: I disagree with the implication that the success of crossbreeding relies entirely on either breeder experience or chance. What about just making sound decisions based on accurate information? Having and using information does not absolutely require experience.
Thank you for your comment. I agree with the point of your comment that citrus breeders should use past data for breeding if available. In fact, we use the past information for breeding. On the other hand, difficulties in preparing and evaluating big populations hampers estimating heritability or QTL analysis in fruit tree. Likewise, preparing populations for test cross or forward and backward crosse is difficult for citrus, and fruit tree. (Please refer to this as an example https://journals.plos.org/plosone/article?id=10.1371%2Fjournal.pone.0162408). As noted in the manuscript, higher heterozygosity of citrus genome make it difficult to predict the trait of a cross in advance. Therefore, the information available for breeding is still quite limited so far (I published these points as a review article recently; please refer [11]). I believe that the above issues are shared among all citrus breeders. For these reasons, I selected this expression.

# “Crossbreeding” is not somatic (asexual) hybridization, but rather it is gametic/sexual hybridization.
Thank you for your comment. I wrote this sentence by referring to references 3 and 10.

# Line 58: Should be “Fruit breeding programs” and there should be background provided of other citrus breeding programs. It is unclear why suddenly the NARO program is described without broader context. The lack of context leads to blurring throughout the rest of the manuscript because it is unclear whether the principles or strategies described are referring to the NARO program or any citrus breeding program. For example, line 206 “The conventional breeding program” – NARO or any program? The heading above the paragraphs describing the NARO program should then mention that program – currently the heading on line 57 is misleading.
Thank you for your comments. I had surveyed the breeding programs conducted by the other groups prior to this review. However, it wasn't easy to figure out a complete picture of each group, as only fragmented information was available. Though that is not the purpose of this review, I suppose it may be meaningful to describe and contrast the breeding strategies of each group. Thus, I revised a sentence in the introduction to indicate this review covers the breeding program in NARO, Japan, and also added new article (new Ref [11]) as for broad reference.

# Line 62, 73, 78, 117, and elsewhere: Not appropriate to refer to “we”/”our” – who is that? Can’t be the team of every author on this manuscript because there is only one author. Avoid such pronouns. The “we” used in line 132 is even more problematic, as it seems to be trying to refer to all citrus breeders or even all conventional breeders of any crop. Also problematic on line 237.

Thank you for your comment. I put this manuscript for proofreading by a native expert twice. The expert did not point out to these sentences, but I rephrased them to clarify the subject.

> L61 To date, the program has developed 39 hybrid varieties with eight intermediary mother selections by repeatedly selecting a superior hybrid from a single cross. When the program started the breeding program, Satsuma, sweet orange, ponkan mandarin, and pummelo were typical in Japan, but each variety had drawbacks.

> L78 Thus, we have continued crossbreeding to develop varieties superior to 'Kiyomi'.
-> Removed this sentence.

> L115 The breeding program provides approximately 5,000 seeds each year for selection and then evaluate them for approximately 20 years; however, only one promising hybrid emerges every few years, and the rest are discarded.

> L129 The citrus breeding program of NARO aims to achieve for satisfying three goals simultaneously: releasing new varieties as quickly as possible, developing new varieties that satisfy varied commercial needs, and improving overall fruit quality.

> L237: Thank you for your comment. I can not find what is problematic in L237, so I did not revise it.

# Lines 64-73, 87-91, and elsewhere should be past tense.
Thank you for your comment. I did not revise L64-73, 87-91 because these sentences describe the feature of those varieties. These traits have not changed in the past, and present.

# Line 72, 86, Table 1: “Brix” (and actually degrees Brix) is a unit of the measurement called SSC (just like “grams”/’g” is a unit of the measurement of fruit size). Therefore, “high Brix content” is incorrect terminology.
Thank you for your comment. I agree to use SSC when we need to indicate a specific value of Brix. On the other hand, when simply referring to sugar content, 'Brix content' is commonly used in citrus research (Please refer [10] as an example). Thus, I did not revise them.

# Line 89: The adjective “good” is vague, non-scientific.
Thank you for your suggestion. This sentence describes the impression of an aroma. As far as I know, there is no quantitative way to describe the aroma. Aroma is always described in intuitive and literary terms even though in the scientific papers except for quantitative analysis (you can find similar expressions in most cited articles). So I use the term 'good' with the understanding that it is a subjective expression, and remained L89.

# Line 113: Should be “alternate”, not “alternative”
Thank you for your suggestion. I corrected “alternative” to “alternative” at L111.

# Lines 126-132: Without any citations of the statements made, it is unclear whether this part is referring to the NARO program or to citrus breeding in general. This paragraph needs referencing to be includable in this review. In fact, this paragraph is the crux of the argument of the manuscript, as it is where is provided the justification for a paradigm shift in citrus breeding. But without any references, it is entirely a “trust me” argument, which is not acceptable for a scientific review paper.
Thank you for your suggestion. I am not sure, but I suppose you may point out L129-130. I revised this sentence. I also cited a reference article at L126-129 (references 3 and renumbered reference 11).

# Line 142: What is “genomic breeding”? Used in the heading but not yet defined nor defined in this section. The word “demand” is probably not appropriate here – something like “opportunity” would fit better.
Thank you for your suggestion. I inserted ‘also simply called genomic-breeding’ at L181-182.

# Line 148-150: Meaning of this statement is unclear. The two parts of the sentence make sense, but it is unclear how they logically connect together.
Thank you for your suggestion. I revised these sentences.

# Line 150-152: This argument needs referencing.
Thank you for your comment. I explained this scheme in Figure 2.

# Line 153: “will dramatically increase the virtual breeding scale” – a strong statement that carries little weight without definition of the terms used here, clearer argumentation, and citation of appropriate studies backing it up.
Thank you for your comment. We have already been evaluating the efficiency of improvement with MAS and GAB. However, it will take additional time to complete the validation in the whole process. Therefore, I am focusing on describing the principle of our strategy in this review. I will report the final results when we have released the varieties selected by MAS/GAB. So I revised these sentences.

> Even if only a few traits are evaluated by MAS, predicting the traits before grafting facilitates increasing the virtual breeding scale and is anticipated to increase the chance of finding promising premium quality hybrids (Figure 2).

# Line 171-188: GWAS and GS are first appropriately defined as two separate things but then they are intermingled from sentence to sentence. The conclusions made on lines 186-189 are therefore not adequately supported. The relevance of each of these two things, separately, to citrus breeding needs to be adequately explained. For example, GS is stated to avoid the concern of segregation of specific trait-associated loci, yet GWAS is stated to detect trait-associated loci. Why not just use GS and avoid GWAS? GWAS is useful towards “conventional” MAS, but not GS. Line 195 further highlights the confusion: “selection by GWAS” does not make sense. In fact, it looks to me that GWAS should not be part of the “2.0” strategy of this manuscript. Lines 432 and 436-7 incorrectly connect the two together immediately as if they are the same thing or as if they easily click in together.
Decision-making on how to use GS or GWAS depends on the target trait and breeding strategy. The number of seedlings to be evaluated, the available budget, and the trait of interest should also be considered. In addition, although not thoroughly examined, the number of genes involved could vary depending on the cross combination and will also affect the use of GS and GWAS.
These are complex matters and there is not enough space to discuss them here, nor is that the purpose of this manuscript. I have previously published two review articles on this issue [11,16], and I cited them in the manuscript for the reader's reference.

# Line 181: “linked to” should be “associated with”. The term “linked”, i.e., regarding genetic linkage, refers to connections between loci – such as marker loci and trait loci. A locus cannot be “linked” to a trait such as fruit size or even a phenotype such as large fruit size – but it can be associated with such.
Thank you for your suggestion. I assume you may point out L171, not L181. I think it is not wrong to use 'link', as mentioned in an article https://www.nature.com/articles/533294d, but I revised it according to your suggestion.

# Line 191: Reword this heading – the grammar has several problems.
I am sorry, but I cannot find the corresponding heading you pointed out around L191.

# Line 196-197: How can fast breeding be a constraint? The constraint is the long juvenile phase, not the proposed solution to overcome it.
I explained why 'fast breeding' becomes a constraint in section 3.1 in detail. Please refer L201-207.

# Line 200: What is the problem with alternate bearing for GAB? Why not use GAB to select against hybrids predicted to have that undesirable phenotype?
Thank you for your comment. The difficulties in quantitative evaluation of alternative bearing are the reason why GAB of this has not been achieved. For the quantitative evaluation of alternative bearing, it is necessary to evaluate the annual variation in fruit set per tree over a long period (typically ten years). However, the fruit set is susceptible to the environment and training method, thining, or else. For analysis of alternative bearing using GS or GWAS, it is mandatory to evaluate a large set of samples with wide genetic variation under uniform growing conditions and uniform management methods for a long period. It may be easy to do in a small plant such as cereal, but this is quite challenging due to large tree size, budget, and labor constraints. Therefore, no reliable alternative bearing study with GWAS or GS has been conducted to date.

# Lines 206-208: I don’t understand the unification made between “existing superior varieties” and “similar varieties”. Just because two varieties are superior does not mean they are genetically similar.
Thank you for your comment. I agree with the point that varieties with similar phenotypes do not necessarily have the same genotype. However, if the genotypes of individuals obtained from the same cross are significantly similar, it is expected that the phenotypes will also resemble each other. This is one of the principles of GS.

# Lines 206-217: Just as for lines 126-132, this paragraph should not be included without more referencing. Every statement of fact should be referenced, while statements of reason should build on the facts in a very logical way that avoids any hint of “trust me, reader”.
I am sorry, but I cannot understand your point. In L208-214, these sentences is not asking "trust me reader", but explain the breeding trilemma by referring the traits of the varieties developed by NARO. These are the facts, not discussing possibilities. Likewise, L214-218 refers to the MAI reported for peach with citations. And then I discussed the problems when applying this idea to citrus. I think these are logical and rationale.

# Line 272-274: The meaning of “chance” here is not clear. Whether there is a high probability of achieving a superior hybrid in a single cross or a low probability, it’s still highly dependent on “chance” because the specific outcomes depend on uncontrolled recombination, segregation, etc. “Chance” (just like “probability”) is a neutral term that requires a qualifier (high, low, etc.).
Do you mean the sentence at L262-264? I use ‘chance’ as a noun, not adjective, to mean opportunity.

# Line 304: “it is known empirically” – by whom? Reference?
Do you mean L288? I appended references (renumbered references 12 and 41).

# Line 360: Unclear why “cytoplasm” is referred to. Should be “maternal parent”. Also, “affected” is not the right word at all, and indicates lack of understanding of inheritance principles. The alleles for any phenotype are inherited from both parents (except in unusual circumstances), so both parents “affect” any trait. Perhaps in this sentence it is trying to refer to the presence and inheritance of dominant-effect alleles.
Do you mean the sentence at L338? No, dominant-effect is not what I wanted to say in this sentence. I recently reported the preferential accumulation of particular cytoplasm type during natural selection (please confirm the renumbered reference [35]). Though it has not been examined in detail, this finding implies that the cytoplasm could affect fruit traits (thus, I explained this with "suggest", not “should be”, in the text at L337). Nevertheless, I revised this sentence to clarify the meaning and to avoid misundestanding.

# Lines 421-430: The argument of this paragraph does not make sense. As stated, rebreeding requires multiple generations. Yet the lead-up to this section is about how citrus breeding does not need multiple generations and especially that would be greatly hampered by and instead requires a solution that bypasses long juvenility. The solution seems to be complementary breeding, in the next paragraph. But how can “a hybrid with improved target traits can be obtained in a short period”? The only evidence provided is in peach-almond that requires multiple generations via MAI.
I am sorry, but I am not sure which statement you are pointing out at L421-430. I assume you may mean L406-418, then rephrased these sentences to explain the complementary breeding as below.

> Complementary breeding initially selects the traits to be improved on a set of seedlings from the same cross of the target variety by MAS with DNA markers developed from GWAS. Then, GS selects the candidate individuals that resemble their traits to the target variety.

# Lines 432 and 436-7: As mentioned previously, the inclusion of GWAS here does not make sense. GWAS is not about selection.
I am sorry, but I am not sure which statement you are pointing out at L432 and 436-7. I assume you may mean L419-420, then appended sentences to explain the mimic breeding as below.

>The procedure of mimic breeding is similar to that of complementary breeding, but mimic breeding does not aim to select individuals close to a particular existing variety for their traits but selects the best candidate individuals that fit the breeding target.

# Line 437: As the manuscript previously describes, GS requires genotyping with genome-wide markers. How is this to be practically and affordably done with “numerous seedlings”?
I am sorry, but I am not sure which statement you pointed out at L437. I assumed you may mean around L426-425. The cost matter is not the main point to be discussed in this review; however, I understand many may be interested in this point. Thus, I inserted a paragraph to introduce how to conduct the GAB with less budget as below at L437.

> The cost is another indispensable matter in conducting GAB. Although the genome-wide genotyping cost has been declining, the expense of genotyping all seedlings is too high to be ignored. However, two-step selection before grafting will contribute to reducing the entire cost of GAB. In the two-step selection, MAS for a large set of seedlings using several DNA markers preliminary selects the primary candidate seedling and then eliminates most of the hopeless seedlings. In this first selection, SSR or SNP markers developed from GWAS are suitable for low-cost selection of a large population. In the following second selection, genome-wide genotypes of the few remaining promising seedlings are obtained, then provide them for GS prediction of fruit traits to select elite individuals. The total cost of citrus breeding to maintain and evaluate many individuals for an extended period is much higher than that of annual crops. Therefore, conducting GAB will suppress the total cost than the cost if only using the phenotypic selection of many seedlings.

# The difference between rebreeding and complementary breeding is not adequately explained. The supposed differences mentioned on lines 431-2 (“but…”) are already components mentioned in rebreeding.
Thank you for the comment. Please refer the reply above.

# Lines 444-448: This paragraph does not adequately describe what mimic breeding is. Nor does it provide any evidence that this is a feasible strategy.
Thank you for the comment. Please refer the reply above.

# Line 451: What does MAS have to do with the success of the approaches? Earlier in the manuscript, MAS was specifically noted as being “conventional” and needing replacement by a new strategy. And what follows in this paragraph is not about MAS anyway, but rather GS.
I am sorry, but I cannot find the corresponding sentence you mentioned around L451 (it is a blank line).

# Lines 456-464: These are all just “trust me” statements, and thus carry little weight.
I am sorry, but I cannot find the corresponding sentence you mentioned at L456-464. Do you mean the legend of Figure 2?

# Lines 468-475: This Conclusion is much too over-reaching. These strong statements are not supported by the rest of the manuscript. There is no evidence provided that the proposed “GAB” strategy has been or will be effective.
I am sorry, but I cannot find the corresponding sentence you mentioned at L468-475. Nevertheless, I revised the Conclusion entirely.

Sincerely, 
Tokurou Shimizu

Reviewer 3 Report

This manuscript present interesting information on the future of citrus breeding. However, looking at the title it seems that it is a general review of citrus breeding programmes. On the contrary, reading the full text, all the information provided seems to belong to the crossbreeding program of the Institute of Fruit Tree and Tea Science, NARO (NIFTS). This should be clearly stated in the title and in the abstract of the paper. You can say that you are using NARO experience to draw conclusions for citrus breeding programmes in general.

Lines 15-17. The sentence “In addition to these efforts, analysis of the genealogy of indigenous citrus varieties revealed that many high-quality indigenous varieties had been selected within several crosses” is not clear. What “several crosses” means?

Lines 79-99 and Figure 1. I do not see the point of the full description of the genealogy of ‘Kiyomi’ cultivar. It is not necessary to support the statements made in the paper on the utility of MAS in future breeding programmes. On the contrary, few information is provided on the steps of selections of the NARO citrus breeding programme and which of those steps will be shorten by the introduction of MAS. Following figure 2 it seems that there are three steps of selection, but the % of genotypes selected in each step with and without MAS is not clear. Maybe you would need very good markers of most of the main traits under selection to really have a difference between conventional and MAS-assisted selection. Following your arguments of lines 169 to 189 this is still far away from practice.

Line 202-203. The main goal of GAB should be to use DNA analysis to perform selection without the need of a phenotypic evaluation. Please explain in more detail why the use of GAB will not drastically reduce the breeding period. In the case GAB is not reducing the breeding period, what is the purpose of use it?

Lines 253-260. It is not necessary to give the names of all the 12+24 parents

Headings 3.3 to 3.7 Please, report if in any of the breeding works reported in those headings, any type of marker assisted selection has been performed.

Reading the conclusions, it seems that GAB is a breeding tool already set up and ready to use. However, giving the current state of the art I think that, up to now, is a “promising strategy” rather than something that could be currently use by the citrus breeding programmes. Authors should clearly separate which molecular techniques are now fully available for citrus breeders and which ones are still “under development”.

Author Response

Dear Reviewer 3,

Thank you for taking the time to review my manuscript, and I sincerely appreciate your kind comment. I am glad to hear your positive comments, and hoping you may found this short review is useful. These comments and suggestions were quite helpful in improving the quality of this manuscript and its readability. I revised the manuscript, and anticipated those changes follow the comments and suggestions. Please confirm the following reply for each suggestion and comment. I would appreciate it if you could confirm the revised manuscript to make it easier to understand and sound.

# This manuscript present interesting information on the future of citrus breeding. However, looking at the title it seems that it is a general review of citrus breeding programmes. On the contrary, reading the full text, all the information provided seems to belong to the crossbreeding program of the Institute of Fruit Tree and Tea Science, NARO (NIFTS). This should be clearly stated in the title and in the abstract of the paper. You can say that you are using NARO experience to draw conclusions for citrus breeding programmes in general.
Thank you for the comment. I appended a sentence that this review covers the breeding program in NARO, Japan. I considered adding NARO or Japan in the title, but it would require a significant title rewriting. Thus, I remained the title without change.

# Lines 15-17. The sentence “In addition to these efforts, analysis of the genealogy of indigenous citrus varieties revealed that many high-quality indigenous varieties had been selected within several crosses” is not clear. What “several crosses” means?
Thank you for the comment. Though "several cross" means "several crossing events", I agree this is umbiguous. I replaced this part with "a few generations" to clarify the meaning.

# Lines 79-99 and Figure 1. I do not see the point of the full description of the genealogy of ‘Kiyomi’ cultivar. It is not necessary to support the statements made in the paper on the utility of MAS in future breeding programmes. On the contrary, few information is provided on the steps of selections of the NARO citrus breeding programme and which of those steps will be shorten by the introduction of MAS. Following figure 2 it seems that there are three steps of selection, but the % of genotypes selected in each step with and without MAS is not clear. Maybe you would need very good markers of most of the main traits under selection to really have a difference between conventional and MAS-assisted selection. Following your arguments of lines 169 to 189 this is still far away from practice.
I included the pedigree of Kiyomi to show how conventional breeding have worked without using GAB. Using this figure, by referring to Table 1, I aim to show the long generation period of the breeding selection, the continuous crossing is necessary to select an elite variety in conventional breeding, and the same varieties are used repeatedly for crossbreeding. These are the background for the motivation to use non-elite varieties for breeding for increasing diversity. Also, MAS has not been used in these processes.
I investigated treeding programs by other research groups beforehand (some of them are introduced in [11]), but showing all breeding strategies in each country is outside the scope of this review. Thus, I focused on the pedigree of Kiyomi in detail in this review. Introducing the past effort of Japanese citrus breeding without misunderstandings is another motivation of this figure. I would appreciate it if you could understand these purposes.

# Line 202-203. The main goal of GAB should be to use DNA analysis to perform selection without the need of a phenotypic evaluation. Please explain in more detail why the use of GAB will not drastically reduce the breeding period. In the case GAB is not reducing the breeding period, what is the purpose of use it?
I agree that the objective of GAB is to avoid phenotypic selection. As mentioned in Section 2, citrus breeding is the selection of superior individual(s) from a single cross (L62-63). In fruit tree and citrus breeding, most of the breeding period consists of the long juvenile period until flowering and the period required for trait evaluation in the field (L158, L189-191). Because both MAS and GAB are the selection method, they do not contribute to shortening the juvenile period. The traits to be selected by GS are quantitative traits involving multiple genes, and the current prediction accuracy is about 80% at maximum.
In addition, traits that MAS or GAB cannot select need to be evaluated in the field. Therefore, even those selected by GAB still need to be evaluated over several years using conventional methods. Due to an upper limit to the number of seedlings that can be maintained in the field, the GAB is advantageous in increasing the probability of promising seedlings emerging by selecting them before grafting (Section 2.3). The above points are illustrated in Figure 2. It takes more than 20 years on average per generation in citrus breeding in NARO (L113-115). Suppose we can shorten the selection of elite seedlings within a single generation, which used to take several generations, by using GAB. In that case, it will be very effective in shortening the breeding period (Section 3.1; L194-196). From my recent analysis of citrus genealogy, I have demonstrated that it is possible to breed varieties of excellent quality even from a single generation of crosses (Section 3). This suggests the possibility of selecting elite candidates in a single generation using GAB selection by referring to the revealed genealogy, even if using a non-elite variety for the breeding parent, contributing to reducing the whole breeding period. The citrus breeding program in NARO is currently working on this idea.

# Lines 253-260. It is not necessary to give the names of all the 12+24 parents
Thank you for your suggestion to improve the quality of the manuscript. I included these names to improve the readability of readers. Although describing these names may seem redundant, I consider it is beneficial to allow readers to identify the corresponding variety names without referring to the cited reference [renumbered reference 35]. For this reason, I decided not to delete these names. I would appreciate it if you could agree this decision.

#Headings 3.3 to 3.7 Please, report if in any of the breeding works reported in those headings, any type of marker assisted selection has been performed.
Please refer to the response to the following comment.

#Reading the conclusions, it seems that GAB is a breeding tool already set up and ready to use. However, giving the current state of the art I think that, up to now, is a “promising strategy” rather than something that could be currently use by the citrus breeding programmes. Authors should clearly separate which molecular techniques are now fully available for citrus breeders and which ones are still “under development”.
MAS and GAB have already been introduced into NARO's citrus breeding program and used for selection for almost ten years. I wrote about the recent advances in MAS in references [11] and [16], so I did not describe them in detail here to avoid redundancy. However, to help the readers understand, I inserted short sentences about currently available DNA markers for selection and their references in the text according to your suggestion (renumbered lines 153-157). 
For your information, we have confirmed this strategy works by selecting promising candidates with MAS/GAB. We are also evaluating the efficiency of improvement with MAS and GAB. However, it will take additional time to complete the validation in the whole process. Therefore, I am focusing on describing the principle of our strategy in this review. I will report the final results when we have released the varieties selected by MAS/GAB.

Sincerely, 
Tokurou Shimizu

Round 2

Reviewer 2 Report

Thank you for your detailed response to each of my concerns and suggestions. Where issues have been sufficiently addressed, either directly or indirectly, I haven’t raised the issue again below. The additional text provided in the manuscript was particularly effective.

Most remaining issues mentioned below are either because of the two of us somehow not having the same manuscript versions with the lines on the same place (so I have had to refer again to these issues for your attention) or because of misunderstandings in what you or I have written, which I assume is a language issue.

Author comment: “I put this manuscript for proofreading by a native expert twice. The expert did not point out to these sentences, but I rephrased them to clarify the subject.

That does not mean the statements (in English language) are now perfect or correct. Revisers often don’t catch every issue. Neither do anonymous peer-reviewers – and so just because a paper is published does not mean it has no mistakes in it (English, conceptual, analytical, etc.). In any case, it is clear that the original manuscript had been well revised prior to submission because the English was generally of very high quality. For me, this attention to quality made it seem worthwhile to me to get into the finer detail in my review(s) and a worthwhile study for other readers to appreciate and build upon scientifically.

Title: Should mention NARO. Reasons: (1) to match with the appropriate qualifying statement added to the end of the Abstract, and (2) see comment below. Also should use the word “proposed” or “suggested” in the title as an adjective to “approach”.

The headings of sections 2 and 2.1 are still problematic. They are generically stated, but what follows, as mentioned in the previous review, focuses just on the NARO program. Because of this insistence on reviewing only to the NARO, I recommend that the manuscript’s title be changed to refer specifically to the NARO program rather than incorrectly implying the strategies proposed are relevant to other citrus programs.

Line 42: “somatic hybridization” – as in my previous review, the meaning of “somatic hybridization” is not consistent with “crossbreeding”, despite other reports the author indicates that make this connection in citrus. Perhaps the author has misinterpreted those previous reports, or else those reports misstated the meaning. In any case, somatic hybridization is NOT gametic/sexual hybridization, but “crossbreeding” IS based on gametic/sexual hybridization.

Line 74’s “Our” still needs fixing. Also on line 232.

Line 89’s “good” still needs fixing. Also line 70. Author’s comment: “I use the term 'good' with the understanding that it is a subjective expression”. But that is your understanding, not the reader’s. As I mentioned last review, the word “good” is vague and non-scientific. If you purposely use it, here in a formal scientific setting, then either put it in quotes (“good”), or define it as much as possible, or explain your understanding that the term is just an impression. Remember, the forum here is formal scientific writing.

“Brix” still needs fixing. Author’s comment: “I agree to use SSC when we need to indicate a specific value of Brix. On the other hand, when simply referring to sugar content, 'Brix content' is commonly used in citrus research (Please refer [10] as an example). Thus, I did not revise them.

The word “Brix” is colloquial, not scientific. In my last review, I carefully explained the reason it is wrong to use the term in scientific writing so that you would be thereafter informed about the reason for the incorrectness. Even if some published reports have used the colloquial term, that does not mean it is right – it just indicates they learned it incorrectly and the mistake was not pointed out to them. There are many words used colloquially, even while we conduct science, that should not then be perpetuated in formal scientific writing. Along those lines, pointing out that someone else has done/written something in a published paper is usually a poor rebuttal and does not mean it is correct. As the saying goes, just because someone jumps off a bridge doesn’t mean you should follow.

Line 150: Still not clear what “virtual breeding scale” means. Similarly, “virtual breeding size” on line 395 is unclear.

GWAS vs. GS (lines 171-189): This section is clear enough. But regarding the author’s rebuttal (“Decision-making on how to use GS or GWAS depends … for the reader’s reference”), the issue is that one of these techniques is NOT a breeding operation. GWAS is research conducted upstream, and easily independently, of breeding. It is about identifying genomic regions associated with traits of interest. While that can be useful for breeding, it is NOT breeding. Same with making a genetic map or sequencing a genome. In contrast, GS is a breeding operation – the “S” stands for “selection” which is an operation that is performed in breeding. This fundamental conceptual difference that was blurred in the original manuscript is the reason for my previous review comments. Indeed, this manuscript is not the place to review these techniques, but it is the place to ensure they are appropriately represented. (By the way, I’m very familiar with all of the genetics concepts and techniques mentioned in this manuscript, especially regarding the peculiarities of citrus and similar crops – so my requests for any clarification should be revisions provided to the readership at large.)

Grammar problem previously mentioned in heading of “Line 191” (now line 186) – this is the heading of Section 3.

“Fast-breeding” being a “constraint”: As I stated in my previous review, “How can fast breeding be a constraint? The constraint is the long juvenile phase, not the proposed solution to overcome it”. I still do not understand how “fast-breeding”, thus the use of short juvenility, is a negative thing for breeding.  Surely fast-breeding is a potential solution! Lines 201 onward that the author refers to in the rebuttal do not explain why fast-breeding or short juvenility are a constraint, a negative thing. It just talks about vague “diversity”. Long juvenility is a constraint, sure, but short juvenility?? Perhaps there is an English problem here rather than a conceptual issue – maybe the author means that achieving the possible solution of fast-breeding is “difficult” rather than “fast-breeding … is another constraint”.

Regarding my previous comment of “What is the problem with alternate bearing for GAB? Why not use GAB to select against hybrids predicted to have that undesirable phenotype?”, the author’s rebuttal explained why it is difficult to identify genomic regions influencing alternate bearing. But this response does not resolve the issue I was pointing out, which is why is alternate bearing even mentioned in this sentence that refers to GAB? As stated (lines 193-5), long juvenility is a constraint to achieving fast-breeding. But how is alternate bearing a constraint to achieving fast-breeding?

Line 203 “crosses between similar varieties” – I made a comment about this part previously. The author’s response just indicated that they didn’t understand my point. I see the issue, though, which is the meaning of “crosses between similar varieties”. By this, I think the author means “repeated crosses between common parents”. Please change to something like this, because the current wording is problematic. As I was trying to point out in my original comment, “similar varieties” reads instead as “similar in phenotype”, especially when following the previous sentence that uses the adjective “superior” for varieties.

My previous comment of “Just as for lines 126-132, this paragraph should not be included without more referencing. Every statement of fact should be referenced, while statements of reason should build on the facts in a very logical way that avoids any hint of “trust me, reader”.” – in the rebuttal, the author mentions that all this refers to the facts of the NARO program. Right! That is what is not yet clear in the paragraph. As I mentioned, more referencing is needed (which would provide that referral to the NARO program) AND, as I’ve pointed out in several other places here and originally, the whole review needs to be much clearer that it is referring to the NARO program and not citrus breeding in general. Once that context is made much clearer (which it is to the author but will not be to the reader until certain changes are made such as to the title and to headings), then yes the statements are logical and rational. As a reviewer, I’m representing the readers at large – something might be clear to the writer, but if it is not clear to a reviewer then you can bet it will not be clear to many readers.

Line 268-269: “highly dependent on chance” – the grammar of this sentence remains incorrect. I explained why last round. Furthermore, “the possibility” is highly dependent on chance” is circular logic, as “possibility” and “chance” mean the same thing in this context. Reword the sentence to better state what the actual point is.

Lines 402-411: Unclear why rebreeding is still mentioned when as described it does not suit citrus. This paragraph does not support the statement on lines 399-401. How is rebreeding now enabled for citrus?

Lines 427-8: “selecting entirely new hybrids by GWAS and GS selection” – the word selection appears three times here! The last one, after GS, should be deleted. Also, as previously mentioned, GWAS is NOT selection, and therefore does not belong in this sentence/approach.

Line 448: “if” should be “of”

My previous comment, unaddressed, of “What does MAS have to do with the success of the approaches? Earlier in the manuscript, MAS was specifically noted as being “conventional” and needing replacement by a new strategy. And what follows in this paragraph is not about MAS anyway, but rather GS.” now refers to line 450 (“The success of these approaches…”). Please address.

My previous comment, unaddressed, of “These are all just “trust me” statements, and thus carry little weight.” now refers to lines 455-463 (“However, enriching … on a simulation.”). More on this in next comment.

Lines 450-463: This paragraph is a good example of how the manuscript still has too many “trust me” statements. It states many times that something “will” happen, and cites no previous scientific studies to substantiate the statements. Nor is the logic strong enough that it is clear the outcome is inevitable. They are interesting ideas, but as presented in this manuscript that’s all they are. In contrast, the revision of the Conclusion has fixed these issues. While the original manuscript’s Conclusion stated that things “will” happen (which is the “trust me” I’m referring to – “trust me, it will happen because I said it will”), instead qualifiers have been added to the Conclusion section such as “support”, “enables”, helpful”, and “anticipated”. These things are expected to happen – you hope they will happen, but you don’t know, and have not provided convincing evidence that they will.

Conclusion (lines 466-475): Only strategies involving planning of effective crosses are mentioned, but the other half of the manuscript’s proposed strategies – involving selection among seedlings – is strangely not mentioned at all here.

Figures 3-4, 5-6, 7-8, 9-10, and 11-12 still contain too much redundancy, given that each pair of figures is the same other than the fruit being of either an external or internal view. Just because a journal has unlimited length opportunity does not mean the reader wants to wade through lots of redundant figures. If a suitable single figure for each pair cannot be created, then simply move one of the paired figures (external or internal view) to Supplementary.

Author Response

Dear Reviewer 2,

Thank you for taking the time to carefully review my manuscript again. I sincerely appreciate your kind comment and suggestions. I revised the manuscript, and anticipated those changes follow the comments and suggestions. Please confirm the following reply for each suggestion and comment.

# Most remaining issues mentioned below are either because of the two of us somehow not having the same manuscript versions with the lines on the same place (so I have had to refer again to these issues for your attention) or because of misunderstandings in what you or I have written, which I assume is a language issue.

I confirmed the line numbers appeared on the right side of each line when I opened the file on MS Office 365. I confirmed the line number in MDPI's template file, too. However, I found that an older version of Word (eg. Word 2013) does not show the line number as anticipated like the latest version of MS Word. I guess that might be a reason for the discrepancy.

# Author comment: “I put this manuscript for proofreading by a native expert twice. The expert did not point out to these sentences, but I rephrased them to clarify the subject.” That does not mean the statements (in English language) are now perfect or correct. Revisers often don’t catch every issue. Neither do anonymous peer-reviewers – and so just because a paper is published does not mean it has no mistakes in it (English, conceptual, analytical, etc.). In any case, it is clear that the original manuscript had been well revised prior to submission because the English was generally of very high quality. For me, this attention to quality made it seem worthwhile to me to get into the finer detail in my review(s) and a worthwhile study for other readers to appreciate and build upon scientifically.

I sincerely appreciate your positive comment.

# The headings of sections 2 and 2.1 are still problematic. They are generically stated, but what follows, as mentioned in the previous review, focuses just on the NARO program. Because of this insistence on reviewing only to the NARO, I recommend that the manuscript’s title be changed to refer specifically to the NARO program rather than incorrectly implying the strategies proposed are relevant to other citrus programs.

Revised to “Past efforts of the citrus breeding program in NARO”.

# Line 42: “somatic hybridization” – as in my previous review, the meaning of “somatic hybridization” is not consistent with “crossbreeding”, despite other reports the author indicates that make this connection in citrus. Perhaps the author has misinterpreted those previous reports, or else those reports misstated the meaning. In any case, somatic hybridization is NOT gametic/sexual hybridization, but “crossbreeding” IS based on gametic/sexual hybridization.

Thank you for the comment. I removed “somatic”.

# Line 74’s “Our” still needs fixing. Also on line 232.

Corrected

# Line 89’s “good” still needs fixing. Also line 70. Author’s comment: “I use the term 'good' with the understanding that it is a subjective expression”. But that is your understanding, not the reader’s. As I mentioned last review, the word “good” is vague and non-scientific. If you purposely use it, here in a formal scientific setting, then either put it in quotes (“good”), or define it as much as possible, or explain your understanding that the term is just an impression. Remember, the forum here is formal scientific writing.

L70, L75: Corrected “good” to “prefarable”.

L89: Enclosed “good” in quotation marks at here.

# “Brix” still needs fixing. Author’s comment: “I agree to use SSC when we need to indicate a specific value of Brix. On the other hand, when simply referring to sugar content, 'Brix content' is commonly used in citrus research (Please refer [10] as an example). Thus, I did not revise them.” The word “Brix” is colloquial, not scientific. In my last review, I carefully explained the reason it is wrong to use the term in scientific writing so that you would be thereafter informed about the reason for the incorrectness. Even if some published reports have used the colloquial term, that does not mean it is right – it just indicates they learned it incorrectly and the mistake was not pointed out to them. There are many words used colloquially, even while we conduct science, that should not then be perpetuated in formal scientific writing. Along those lines, pointing out that someone else has done/written something in a published paper is usually a poor rebuttal and does not mean it is correct. As the saying goes, just because someone jumps off a bridge doesn’t mean you should follow.

Corrected

# Line 150: Still not clear what “virtual breeding scale” means. Similarly, “virtual breeding size” on line 395 is unclear.

L151: I revised the section 2.3 thoroughly to define and explain about "virtual breeding size".

L396: Not revised “virtual breeding size” since it was described at L151.

# Line 89: The adjective “good” is vague, non-scientific.

Thank you for your suggestion. This sentence describes the impression of an aroma. As far as I know, there is no quantitative way to describe the aroma. Aroma is always described in intuitive and literary terms even though in the scientific papers except for quantitative analysis (you can find similar expressions in most cited articles). So I use the term 'good' with the understanding that it is a subjective expression, and remained L89.

# GWAS vs. GS (lines 171-189): This section is clear enough. But regarding the author’s rebuttal (“Decision-making on how to use GS or GWAS depends … for the reader’s reference”), the issue is that one of these techniques is NOT a breeding operation. GWAS is research conducted upstream, and easily independently, of breeding. It is about identifying genomic regions associated with traits of interest. While that can be useful for breeding, it is NOT breeding. Same with making a genetic map or sequencing a genome. In contrast, GS is a breeding operation – the “S” stands for “selection” which is an operation that is performed in breeding. This fundamental conceptual difference that was blurred in the original manuscript is the reason for my previous review comments. Indeed, this manuscript is not the place to review these techniques, but it is the place to ensure they are appropriately represented. (By the way, I’m very familiar with all of the genetics concepts and techniques mentioned in this manuscript, especially regarding the peculiarities of citrus and similar crops – so my requests for any clarification should be revisions provided to the readership at large.)

Thank you for the advice. I suppose the reviewer mentions about a paragraph at L166-184 in section 2.3. I revised this section 2.3 thoroughly.

# Grammar problem previously mentioned in heading of “Line 191” (now line 186) – this is the heading of Section 3.

Rephrased to “Suggestions from the citrus genealogy for breeding”.

# “Fast-breeding” being a “constraint”: As I stated in my previous review, “How can fast breeding be a constraint? The constraint is the long juvenile phase, not the proposed solution to overcome it”. I still do not understand how “fast-breeding”, thus the use of short juvenility, is a negative thing for breeding.  Surely fast-breeding is a potential solution! Lines 201 onward that the author refers to in the rebuttal do not explain why fast-breeding or short juvenility are a constraint, a negative thing. It just talks about vague “diversity”. Long juvenility is a constraint, sure, but short juvenility?? Perhaps there is an English problem here rather than a conceptual issue – maybe the author means that achieving the possible solution of fast-breeding is “difficult” rather than “fast-breeding … is another constraint”.

I thoroughly revised section 3.1 (L194-196) along with other mentioned points. In section 3.1, I added paragraphs to introduce the previous study that is part of this approach's basis (references 31 and 32). I didn't describe those points in my previous manuscript because the contents had already been discussed in the cited reference 13, for avoiding duplication. However, I rearranged and described it in this manuscript to clarify the background and rationale in this review article.

# Regarding my previous comment of “What is the problem with alternate bearing for GAB? Why not use GAB to select against hybrids predicted to have that undesirable phenotype?”, the author’s rebuttal explained why it is difficult to identify genomic regions influencing alternate bearing. But this response does not resolve the issue I was pointing out, which is why is alternate bearing even mentioned in this sentence that refers to GAB? As stated (lines 193-5), long juvenility is a constraint to achieving fast-breeding. But how is alternate bearing a constraint to achieving fast-breeding?

I thoroughly revised section 3.1 (L194-196) along with other mentioned points.

# Line 203 “crosses between similar varieties” – I made a comment about this part previously. The author’s response just indicated that they didn’t understand my point. I see the issue, though, which is the meaning of “crosses between similar varieties”. By this, I think the author means “repeated crosses between common parents”. Please change to something like this, because the current wording is problematic. As I was trying to point out in my original comment, “similar varieties” reads instead as “similar in phenotype”, especially when following the previous sentence that uses the adjective “superior” for varieties.

I revised this section thoroughly.

# My previous comment of “Just as for lines 126-132, this paragraph should not be included without more referencing. Every statement of fact should be referenced, while statements of reason should build on the facts in a very logical way that avoids any hint of “trust me, reader”.” – in the rebuttal, the author mentions that all this refers to the facts of the NARO program. Right! That is what is not yet clear in the paragraph. As I mentioned, more referencing is needed (which would provide that referral to the NARO program) AND, as I’ve pointed out in several other places here and originally, the whole review needs to be much clearer that it is referring to the NARO program and not citrus breeding in general. Once that context is made much clearer (which it is to the author but will not be to the reader until certain changes are made such as to the title and to headings), then yes the statements are logical and rational. As a reviewer, I’m representing the readers at large – something might be clear to the writer, but if it is not clear to a reviewer then you can bet it will not be clear to many readers.

I revised section 2.2 and also revised sections 2.3, and 2.4 thoroughly. In the revision of section 2.4, I introduced previous studies of the segregation analysis of Kiyomi that support this approach. These points have already been discussed in the cited article [13], and I had not mentioned them in the previous submission to avoid duplication. However, according to the suggestions, I rearranged the content and added new sentences to present the rationale.

# Line 268-269: “highly dependent on chance” – the grammar of this sentence remains incorrect. I explained why last round. Furthermore, “the possibility” is highly dependent on chance” is circular logic, as “possibility” and “chance” mean the same thing in this context. Reword the sentence to better state what the actual point is.

Revised to “Therefore, with the current systematic breeding program, the possibility of selecting an in-dividual with traits comparable to those of Satsuma is insignificant.”

# Lines 402-411: Unclear why rebreeding is still mentioned when as described it does not suit citrus. This paragraph does not support the statement on lines 399-401. How is rebreeding now enabled for citrus?

I revised this section.

# Line 448: “if” should be “of”

Corrected.

# My previous comment, unaddressed, of “What does MAS have to do with the success of the approaches? Earlier in the manuscript, MAS was specifically noted as being “conventional” and needing replacement by a new strategy. And what follows in this paragraph is not about MAS anyway, but rather GS.” now refers to line 450 (“The success of these approaches…”). Please address.

I revised L450-451 to clarify the meaning.

“The success of these approaches relies on the accuracy of selection, and the variation of selectable traits by MAS or GS. The cross combination is another factor that affects the probability of selecting a target individual.”

# My previous comment, unaddressed, of “These are all just “trust me” statements, and thus carry little weight.” now refers to lines 455-463 (“However, enriching … on a simulation.”). More on this in next comment.

# Lines 450-463: This paragraph is a good example of how the manuscript still has too many “trust me” statements. It states many times that something “will” happen, and cites no previous scientific studies to substantiate the statements. Nor is the logic strong enough that it is clear the outcome is inevitable. They are interesting ideas, but as presented in this manuscript that’s all they are. In contrast, the revision of the Conclusion has fixed these issues. While the original manuscript’s Conclusion stated that things “will” happen (which is the “trust me” I’m referring to – “trust me, it will happen because I said it will”), instead qualifiers have been added to the Conclusion section such as “support”, “enables”, helpful”, and “anticipated”. These things are expected to happen – you hope they will happen, but you don’t know, and have not provided convincing evidence that they will.

Thank you for the comment. I added sentences in this paragraph (L450-463) to describe the bases of these points with references, and then revised the following sentences.

# Conclusion (lines 466-475): Only strategies involving planning of effective crosses are mentioned, but the other half of the manuscript’s proposed strategies – involving selection among seedlings – is strangely not mentioned at all here.

Thank you for your comment. I revised this part.

# Figures 3-4, 5-6, 7-8, 9-10, and 11-12 still contain too much redundancy, given that each pair of figures is the same other than the fruit being of either an external or internal view. Just because a journal has unlimited length opportunity does not mean the reader wants to wade through lots of redundant figures. If a suitable single figure for each pair cannot be created, then simply move one of the paired figures (external or internal view) to Supplementary.

I consider these figures are informative for both breeders and scientist of citrus, but revised to move the cross section figures for supplementals as suggested.

Sincerely,

Tokurou Shimizu

              Tokurou Shimizu, Ph.D.

              Citrus Research Division, Institute of Fruit Tree and Tea Science, NARO

              Okitsu 485-6, Shimizu, Shizuoka 424-0292, Japan

              TEL:      +81 54 369 7108

              E-mail:  tshimizu@affrc.go.jp

Reviewer 3 Report

Respecto to my comments about sentence in # Line 202-203, I found the authors explanation fully convincing. However, to make the paper fully understandable, I would ask authors to include in the manuscript some of the arguments gave in that explanation.

Similarly, in their reply to my comments on the conclusions, in the final paragraph starting by "For your information.." authors are giving valuable information that I think should be included in the mansucript.

Author Response

Dear Reviewer 3,

Thank you for taking the time to review my manuscript again. I sincerely appreciate your helpful comment. I have revised the manuscript along with the comments from another reviewer. I hope those changes meet your suggestions.

# Respecto to my comments about sentence in # Line 202-203, I found the authors explanation fully convincing. However, to make the paper fully understandable, I would ask authors to include in the manuscript some of the arguments gave in that explanation.

Thank you for the comments and suggestions. I combined my previous comments into sections 2.3 and 3.1, then revised these sections thoroughly. In section 3.1, I added paragraphs to introduce the previous study that is part of this approach's basis (references 31 and 32). I didn't describe those points in my previous manuscript because the contents had already been discussed in the cited reference 13, for avoiding duplication. However, I rearranged and described it in this manuscript to clarify the background and rationale in this review article.

# Similarly, in their reply to my comments on the conclusions, in the final paragraph starting by "For your information.." authors are giving valuable information that I think should be included in the mansucript.

Thank you for the suggestion. I added those sentences in Section 5.

Sincerely,

Tokurou Shimizu

              Tokurou Shimizu, Ph.D.

              Citrus Research Division, Institute of Fruit Tree and Tea Science, NARO

              Okitsu 485-6, Shimizu, Shizuoka 424-0292, Japan

              TEL:      +81 54 369 7108

              E-mail:  tshimizu@affrc.go.jp
